# A tough nitric oxide-eluting hydrogel coating suppresses neointimal hyperplasia on vascular stent

Yin Chen [1,2,5], Peng Gao [3,5], Lu Huang [1], Xing Tan [3], Ningling Zhou [3], Tong Yang [3], Hua Qiu [3], Xin Dai [2], Sean Michael [2], Qiufen Tu [3], Nan Huang [3], Zhihong Guo [2], Jianhua Zhou [1,4 ✉], Zhilu Yang [3 ✉] & Hongkai Wu [2 ✉]

Vascular stent is viewed as one of the greatest advancements in interventional cardiology. However, current approved stents suffer from in-stent restenosis associated with neointimal hyperplasia or stent thrombosis. Herein, we develop a nitric oxide-eluting (NOE) hydrogel coating for vascular stents inspired by the biological functions of nitric oxide for cardiovascular system. Our NOE hydrogel is mechanically tough and could selectively facilitate the adhesion of endothelial cells. Besides, it is non-thrombotic and capable of inhibiting smooth muscle cells. Transcriptome analysis unravels the NOE hydrogel could modulate the inflammatory response and induce the relaxation of smooth muscle cells. In vivo study further demonstrates vascular stents coated with it promote rapid restoration of native endothelium, and persistently suppress inflammation and neointimal hyperplasia in both leporine and swine models. We expect such NOE hydrogel will open an avenue to the surface engineering of vascular implants for better clinical outcomes.

---

[1] School of Biomedical Engineering, Sun Yat-sen University, Shenzhen 518107, China. [2] Department of Chemistry, The Hong Kong University of Science and Technology, Hong Kong, China. [3] Key Laboratory of Advanced Technologies of Materials, Ministry of Education, School of Materials Science and Engineering, Southwest Jiaotong University, Chengdu 610031, China. [4] Division of Engineering in Medicine, Brigham and Women's Hospital, Harvard Medical School, Cambridge, MA 02139, USA. [5] These authors contributed equally: Yin Chen, Peng Gao. ✉email: zhoujh33@mail.sysu.edu.cn; zhiluyang1029@swjtu.edu.cn; chhkwu@ust.hk

Vascular stent, which is implanted into a narrowed blood vessel through guided balloon dilation, is regarded as the most effective means for treating coronary artery disease[1]. Since its introduction in the 1980s, vascular stent has been widely employed in interventional cardiology. Compared with the earlier plain balloon angioplasty, the use of first-generation bare-metal stents (BMSs) has already presented remarkable benefits in terms of less acute vessel closure and constrictive remodeling[2]. Despite these advantages, the drawbacks of BMSs were soon reported, including acute inflammation elicited by foreign-body reaction and in-stent restenosis (ISR) induced by neointimal hyperplasia (NIH)[3]. As an alternative, drug-eluting stents (DESs) with a polymer coating carrying anti-cell-proliferative drugs were developed and became the standard of care in percutaneous coronary intervention (PCI)[4]. Although DES has successfully alleviated inflammation and dramatically reduced the rate of early ISR, the released drugs also suppress endothelial cells, thereby increasing the risk of late NIH and stent thrombosis due to impaired endothelialization[5].

To address these complications associated with vascular stent, it is advisable to learn from nature. The inner lining of blood vessel is a monolayer of tightly connected endothelial cells called as endothelium[6]. Native endothelium generates versatile biomolecules to regulate cardiovascular homeostasis and maintain the patency of blood vessel[7]. Among these biomolecules, nitric oxide (NO) plays pleiotropic roles such as prevention of thrombosis, regulation of vasomotion, promotion of endothelial regeneration, and modulation of inflammatory response[7]. As a result, NO is the most widely investigated signaling molecule in the cardiovascular system.

With the knowledge in the biological functions of NO, we envisaged that a coating with sustained release of NO at physiological level might solve the issue of ISR for vascular stents. Such coating must also provide a microniche favored by endothelial cells, so that native endothelium could rapidly form to replace its biological functions. To achieve this goal, hydrogels seem to be the best candidate compared with direct surface engineering, polymeric coatings, and ceramic films because of several reasons. First, they are three-dimensionally (3D) cross-linked aqueous materials with the best resemblance to native tissues[8,9]. Second, they allow for versatile physical and chemical modifications for special purposes. Last but not least, as bulk materials, they can be tailored as highly efficient carriers for therapeutics. In fact, many hydrogels have been exploited as carriers of drugs[10,11], biomolecules[12,13], or cells[14–16] for various applications. However, developing a hydrogel coating for vascular stents is challenging because it must be biocompatible, endothelial cell-adhesive, convenient for handling, and mechanically strong to withstand balloon dilation during angioplasty. Unfortunately, few of the hydrogels reported so far met all these requirements.

In this work, we develop a nitric oxide-eluting (NOE) hydrogel coating for vascular stents. The hydrogel is primarily composed of alginate and gelatin that are analogous to hyaluronic acid and collagen in the extracellular matrix (ECM). This combination could facilitate the selective adhesion of endothelial cells. By tuning the proportions of these two biopolymers and the interaction between them, it becomes mechanically tough. Through conjugation of an organoselenium species to alginate, the hydrogel is capable of persistently catalyzing the generation of NO. Such NOE hydrogel coating can withstand the balloon dilation during angioplasty, prevent thrombosis, inhibit the overgrowth of smooth muscles, promote the rapid restoration of native endothelium, and effectively suppress NIH on vascular stents.

## Results

**Design, synthesis, and optimization of the hydrogel.** To generate a uniform hydrogel coating on vascular stents, a good strategy is in situ gelation. One possible approach to achieving this is dip-coating the stents with the hydrogel precursor solution that cures subsequently. At first glance, it seems the sol–gel transition of gelatin solution can be utilized for that purpose, which liquefies when the temperature is above the melting point ($T_m$) of gelatin and solidifies after cooling. However, the $T_m$ of gelatin (~30 °C) is much lower than that of our body temperature (37 °C)[17], which suggests that chemical cross-linking needs to be introduced in the hydrogel. The reaction of such chemical cross-linking must be cytocompatible and mild. Considering cytocompatibility, Michael-addition reaction is one of the best candidates as it produces no by-products[18]. However, conventional thiol-maleimide addition proceeds too fast at physiological conditions[19], which is inconvenient for manipulation. In our previous work, we reported that Michael-addition reaction between amine and maleimide could be leveraged for cross-linking a hydrogel[20,21]. Compared with thiol–maleimide addition, the amine–maleimide addition is milder, allowing for more maneuverability during the preparation of hydrogel. Based on our findings, we envisioned that a hybrid hydrogel formed by the cross-linking between maleimide-modified alginate (maleimidyl alginate, A–M for short) and gelatin could be tailored as the NOE hydrogel coating for vascular stents (Fig. 1a).

A–M was synthesized by the coupling reaction between pristine alginate and N-(2-aminoethyl)maleimide trifluoroacetate (AEM.TFA) (Supplementary Fig. 1). By tuning the feed ratio between them, A–Ms with varying degrees of modification (DM) were obtained. In our nomenclature, A–M($x$) indicates that $x$% structural units in alginate are coupled by maleimide. In total, three variants of A–M were prepared, including A–M(4.8), A–M(9.3), and A–M(15.9) (see [1]H-NMR, Supplementary Fig. 2). In a preliminary trial, we mixed the precursor solutions (10 w/v%, pH~7.5) of A–M(9.3) and gelatin (G) at different mass ratios and cured them at 37 °C. The gelation processes of them were investigated by measuring the changes of their shear moduli at 37 °C. As shown in Supplementary Fig. 3, although these hydrogels had different stiffness, all of them exhibited similar cure kinetics and reached full mechanical strength within 36 h. To further enhance their strength, we increased the mass concentration of the precursor solutions to 15 w/v%. This time, all three variants of A–M and pristine alginate were mixed with gelatin. A prolonged gelation time (72 h) was assumed to ensure that all hydrogels were fully cured.

As anticipated, pure gelatin was unable to maintain solid state and pristine alginate could not form a hybrid hydrogel with it at 37 °C (Fig. 1b). With the use of A–M, hybrid hydrogels were successfully generated. Notably, gelation only occurred when the proportions of A–M and gelatin were in proper ranges. A–M with a higher DM tended to form stiffer hydrogels with gelatin in a broader range. In addition, the higher the DM of A–M, the more gelatin was required to achieve an optimum cross-linking density. Subsequently, we selected the hydrogels with shear moduli beyond 1.0 kPa at 37 °C (encompassed by the dashed framework in Fig. 1b) for further investigation. Additional dynamic mechanical test (Fig. 1c) suggests that the hydrogels are much stronger at ambient temperature (~25 °C), which is in accordance with our intuition as gelatin itself forms a physical hydrogel below its $T_m$. The ratios of shear moduli at 37 °C and 25 °C ($G_{37\,°C}/G_{25\,°C}$) lie between 0.19 and 0.48, and A–M/G hydrogel with a higher content of A–M is consistently more refractory to softening at 37 °C, given that the DM of A–M is constant.

To further unravel the mechanical behaviors of these hydrogels, tensile testing was conducted. The stress–strain curves (Fig. 1d) of them are diverse that some hydrogels are weak, while others are strong and can tolerate high strain (Supplementary Video 1). Nonetheless, stiffening at higher strain is common in all groups. Quantitative assessment (Supplementary Fig. 4) indicates that the Young's modulus, fracture strength, fracture strain, and

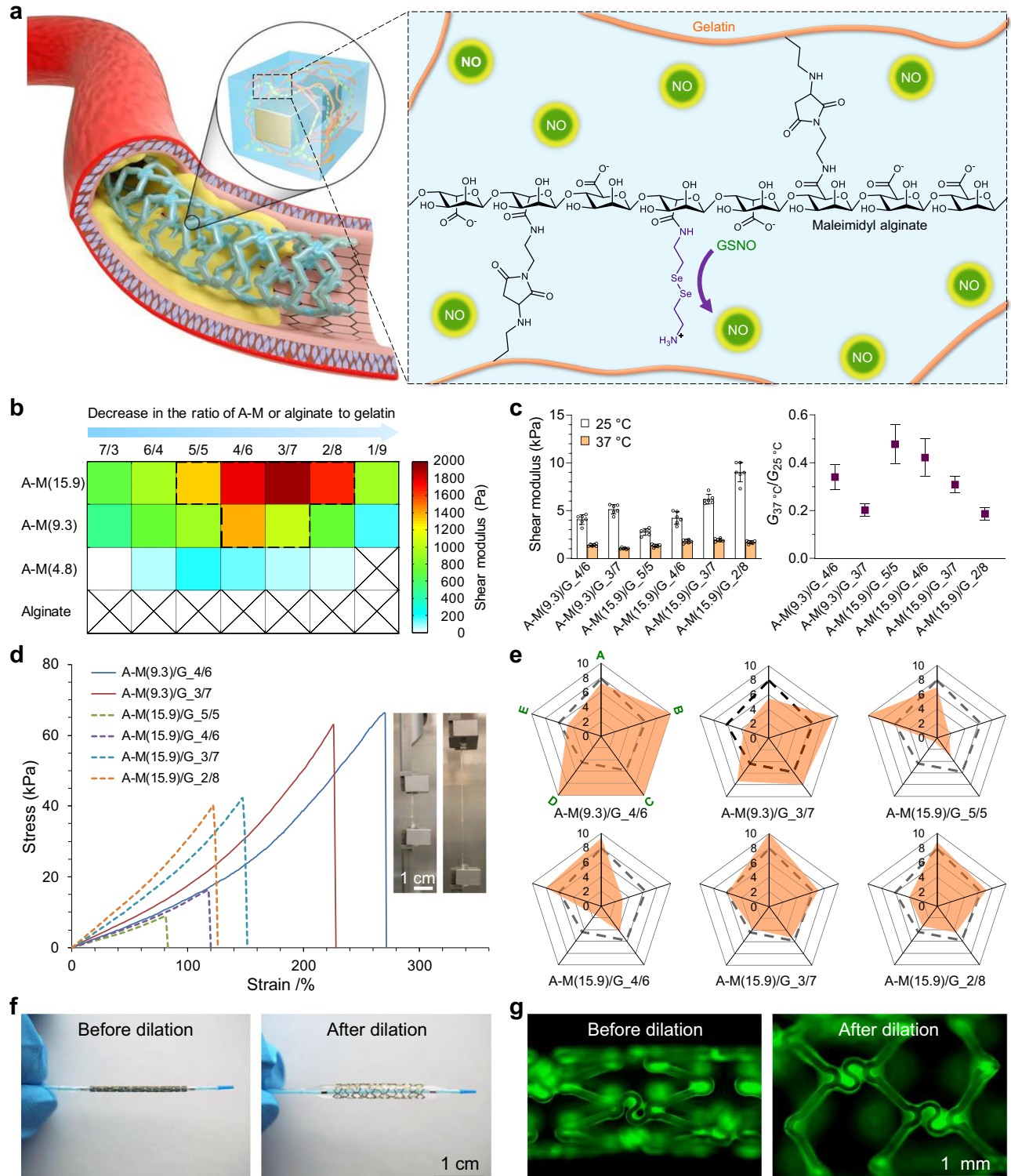

**Fig. 1 Development of a tough nitric oxide-eluting (NOE) hydrogel coating for vascular stent. a** Schematic for the design of our NOE hydrogel. **b** Shear moduli (at 37 °C) of the hybrid hydrogels formulated with maleimidyl alginate (A–M) of varying degrees of modification (DM) and gelatin (G) at different mass ratios. **c** Comparison in shear modulus among the selected hydrogels at 25 °C and 37 °C (mean ± SD, $n = 6$ independent samples). **d** Tensile testing of them at ambient temperature. The insets exhibit the photographs of A–M(9.3)/G_4/6 hydrogel prior to and during extension. **e** Radar charts showing their scores in shear modulus (at 37 °C) (A), fracture strength (B), fracture strain (C), toughness (D), and capacity for further modification (E). Dashed frameworks represent the average values. **f** Photographs of a vascular stent coated with A–M(9.3)/G_4/6 hydrogel before and after balloon dilation in PBS at 37 °C. **g** Fluorescence images of it before and after balloon dilation. The hydrogel coating was labeled with fluorescein isocyanate.

toughness of them are in the ranges of [12.4, 27.3] kPa, [10.4, 60.7] kPa, [84.3, 271.4]%, and [4.2, 69.2] kJ m$^{-3}$, respectively. For comparison, a comprehensive scoring system concerning the mechanical properties of the hydrogels and their capacity for further modification was established (Fig. 1e). Among them, A–M(9.3)/G_4/6 demonstrates the best overall performances. In particular, this hydrogel is remarkably strong and flexible to be consecutively bent, twisted, and knotted without damage at ambient temperature (Supplementary Fig. 5). Consequently, we selected it as the base material to make our NOE hydrogel coating for vascular stents. BMSs of 316 L stainless steel (SS) were assumed as the platform for our initial attempt. Since stainless steel contains no amino groups, a thin film of poly(dopamine-co-hexanediamine) (P(DA-co-HDA))[22] was predeposited onto the BMSs to facilitate their bonding. To assess the strength of the hydrogel on the stent, we coated it with A–M(9.3)/G_4/6 and simulated the process of angioplasty (Fig. 1f) in phosphate-buffered saline (PBS, pH 7.4) at 37 °C. Fortunately, no clear damage was identified in the hydrogel coating (Fig. 1g; Supplementary Fig. 6 to Fig. 8) even if it had endured a very high pressure (up to 8 MPa) for one minute. Such preliminary result was promising and warranted further investigation.

**Mechanism on the toughness of the hydrogel.** Conventional hydrogels are normally weak because they are cross-linked by pure chemical bonds or physical interactions. However, some of our A–M/G hydrogels presented good mechanical properties at ambient temperature. In particular, A–M(9.3)/G_4/6 could be extended by nearly three times with a stiffness comparable to that of muscles[23]. Interpreting such character is important for the application of our materials and the design of hydrogels with better mechanical property. Extensive studies have shown that a highly stretchable hydrogel generally has some mechanism to dissipate the energy built up during its deformation. In our case, we hypothesized that the physical interaction within gelatin or between gelatin and A–M was responsible for energy dissipation. To unravel this, we started our investigation by altering the chemistry of gelatin to obtain gelatin glycinamidate (GelGA or GG in short) and gelatin methacrylate (GelMA or GM in short) (Supplementary Fig. 9 and 10).

Gelatin exists as random coils in an aqueous solution above its $T_m$[24]. The solution spontaneously transforms into a hydrogel when it cools. At the same time, the random coils bind to form ordered triple helices that function as physical cross-links for hydrogel formation[24]. This coil–helix transition is reversible and readily affected by many factors, including pH, ionic strength, and chemical modification to gelatin[25]. Circular dichroism (CD) was employed to study the influence of chemical modification to gelatin on the molecular structure of its hydrogel at ambient temperature. The CD spectra show a strong negative peak around 240 nm (Fig. 2a) that is assigned to the triple helices of pristine gelatin hydrogel. The same peak is also found for GelGA hydrogel, but the intensity of it decreases with a 2-nm blue shift, implying the impairment of the triple helices. For GelMA hydrogel, the peak is almost gone, indicating the random coils predominate in it. In addition, the shear modulus of pristine gelatin hydrogel at ambient temperature is 8.9 ± 0.6 kPa, but it significantly declines to 3.3 ± 0.3 kPa ($P < 0.0001$) and 1.3 ± 0.4 kPa ($P < 0.0001$) for GelGA and GelMA hydrogels, respectively (Supplementary Fig. 11a). Tensile testing (Fig. 2a and Supplementary Fig. 12) also disclosed the similar trend. Moreover, the $T_m$ of gelatin hydrogel is 30.6 ± 0.2 °C, but it decreases to 29.4 ± 0.2 °C and 27.9 ± 0.3 °C for GelGA and GelMA hydrogels, respectively (Fig. 2a and Supplementary Fig. 11b). Taken together, all information reflects that the physical interaction within gelatin hydrogel was destroyed by the chemical modifications.

On top of these findings, we further mixed A–M(9.3) with GelGA or GelMA to make A–M(9.3)/GG_4/6 and A–M(9.3)/GM_4/6 hydrogels. As a negative control, we also prepared alginate/gelatin (A/G_4/6) hydrogel. Tensile testing was conducted on these hydrogels, except for A–M(9.3)/GM_4/6, which was too brittle and repeatedly broke during demolding. Indeed, this hydrogel transformed into a liquid upon heating above the $T_m$ of GelMA, suggesting no chemical cross-link had formed in it. Such phenomenon verified the necessity of amino groups for the chemical cross-linking of our hydrogel since all of them had been amidated in GelMA (Supplementary Fig. 10). The mechanical strength of A/G_4/6 is also very weak and almost identical to that of pristine gelatin hydrogel (Supplementary Fig. 12 and 13). In fact, no chemical cross-links exist in A/G_4/6 as well since pristine alginate does not contain any maleimidyl group. The fracture strength, fracture strain and toughness of A/G_4/6 are merely 11.2%, 19.1% and 3.8% as those of A–M(9.3)/G_4/6 (Fig. 2b and Supplementary Fig. 13). The difference in tensile behavior between A/G_4/6 and A–M(9.3)/G_4/6 manifests the importance of chemical cross-linking in the mechanical strength of the hydrogels. Nevertheless, this does weaken the role of physical interaction, as the mechanical strength of A–M(9.3)/GG_4/6 is much lower than that of A–M(9.3)/G_4/6 even if the triple-helix structure of GelGA is only slightly impaired. We believe both interactions are important for producing a tough hydrogel. To verify this point, we compared the toughness of A–M(9.3)/G_4/6, gelatin, and photo-cross-linked GelMA on metal springs by performing 1000 cycles of stretching and compressing. Our result (Supplementary Fig. 14) demonstrated that only A–M(9.3)/G_4/6 preserved integrity after the test.

Based on the outcomes above, we can rationally deduce the mechanism on the toughness of A–M(9.3)/G_4/6 (Fig. 2c). When a tensile force is exerted on this hybrid hydrogel, it is gelatin that feels the stress in the first place due to its 3D network cross-linked through the triple helices. The stress will quickly propagate to the chemical cross-links between A–M(9.3) and gelatin upon the full stretch of the random coils within gelatin. At this stage, more and more chemical cross-links are involved, bringing about the stiffening of the hydrogel. In the meantime, the triple helices undergo disassembly, dissipating the energy built up during deformation until the fracture of the hydrogel. Thanks to the synergism between physical interaction and chemical cross-linking, A–M(9.3)/G_4/6_hydrogel is mechanically tough.

**Catalytic generation of NO in the hydrogel.** In the cardiovascular system, two pathways have been discovered for the production of NO. One is through endothelial nitric oxide synthase (eNOS), which catalyzes the degradation of L-arginine into NO[26]. The other is through glutathione peroxidase-3 (GPx-3), which metabolizes endogenous nitrosated thiols (RSNO) to generate NO[27]. The pathway of eNOS is complex and involves many components, so that harnessing it is difficult. In contrast, NO generation catalyzed by GPx-3 is relatively simple. The selenocysteine residue of this enzyme is believed to be the catalytic center[28]. In fact, many selenium species have shown the capacity to catalyze the degradation of RSNO into NO. For instance, selenocystamine (SeCA) is capable of catalyzing the production of NO in a mechanism proposed by Meyerhoff et al.[29] (Fig. 3a). This pathway can be readily exploited, so that we made use of it in our NOE hydrogel by conjugating SeCA to A–M(9.3). Inductively coupled plasma mass spectrometry (ICP-MS, Supplementary Fig. 15a) disclosed that the content of conjugated SeCA was about 0.016 mmol g$^{-1}$ in it. By tuning the proportions of SeCA-conjugated A–M(9.3) and blank A–M(9.3), NOE hydrogels conjugated with varying contents (0.2–1.0 mM) of SeCA were prepared.

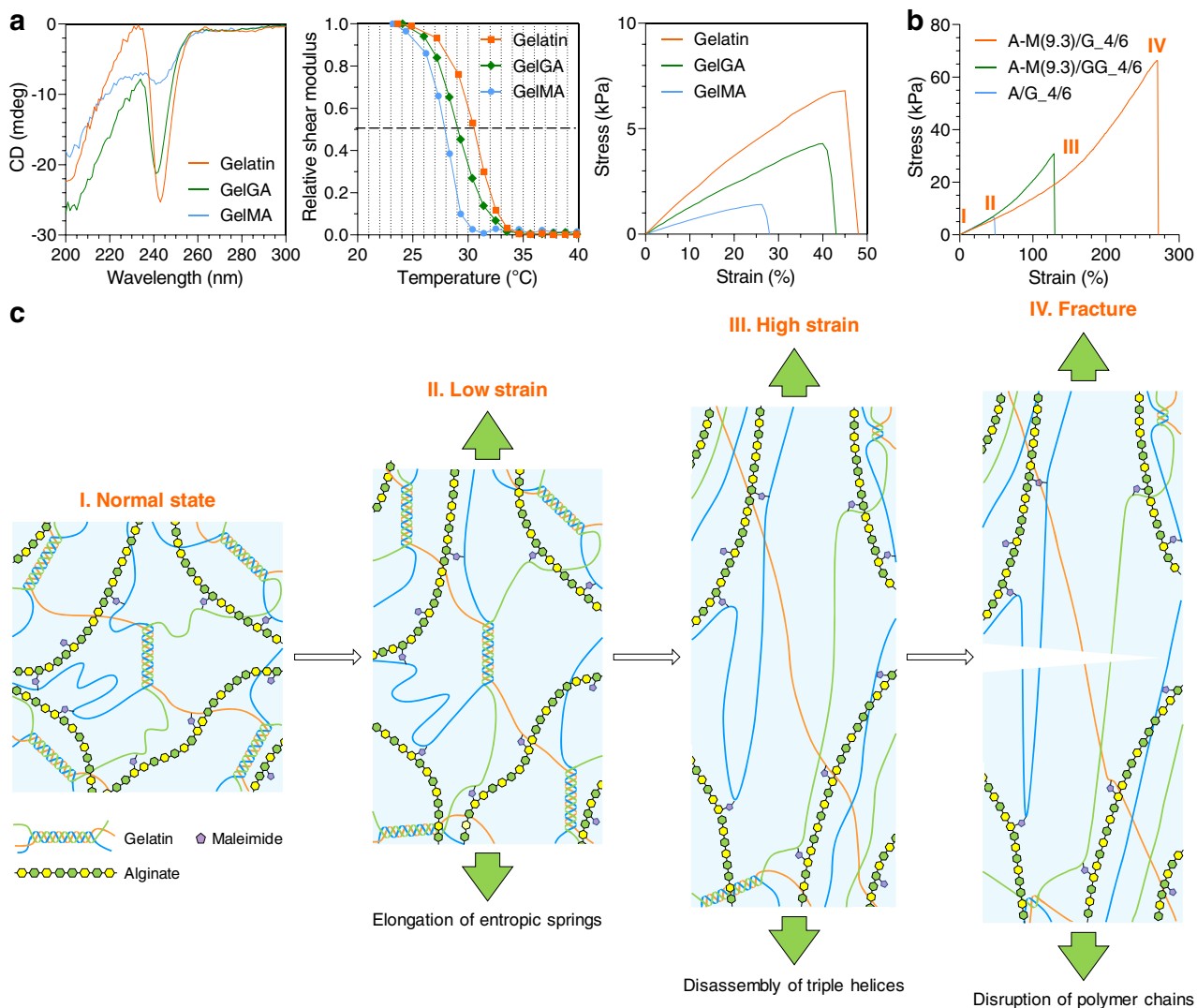

**Fig. 2 Mechanism on the toughness of A–M(9.3)/G_4/6 hydrogel. a** Effects of different chemical modifications to gelatin on the molecular structure, melting point, and tensile behavior of its hydrogel. **b** Comparison in tensile behavior among A-M(9.3)/G_4/6, A-M(9.3)/GG_4/6, and A/G_4/6 hydrogels. **c** Schematic illustration for the mechanism on the toughness of A–M(9.3)/G_4/6 hydrogel.

In most studies, the flux of NO, defined as the production of it per unit area per unit time, was measured for a coating. Since the weight of an NOE hydrogel on a substrate might vary significantly, we assumed the release rate of NO (production of NO per unit mass per unit time) to accurately reflect the NO-generating capacity of it. We detected the release rates of NO from S-nitrosoglutathione (GSNO, 10 μM in PBS at 37 °C) catalyzed by the NOE hydrogels (Fig. 3b). As anticipated, the blank hydrogel is unable to catalyze the generation of NO from GSNO. With the conjugation of SeCA, a burst release of NO followed by gradual decline until steady state was observed. To our delight, the flux of NO was nearly proportional to the release rate of it (Fig. 3c), implying the uniform coating density of the NOE hydrogels on the substrates. At the average coating density of 22.9 mg cm$^{-2}$, the mean flux of NO catalyzed by the NOE hydrogel containing 1.0 mM SeCA was $6.19 \times 10^{-10}$ mol cm$^{-2}$ min$^{-1}$, approaching the normal level from native endothelium[30]. Though the steady release rate of NO is proportional to the content of conjugated SeCA, the peak value presents a quadratic relationship with it (Fig. 3d), leading to the highest burst-release ratio of NO ($3.48 \pm 0.92$) in the NOE hydrogel containing 1.0 mM SeCA (Supplementary Fig. 15b). Nevertheless, this value is still much smaller than those of many other NO-eluting coatings[31]. The

transient burst of NO ($26.3 \pm 4.2 \times 10^{-10}$ mol cm$^{-2}$ min$^{-1}$ at maximum) is unlikely to be detrimental for endothelial cells as it only lasts for a few minutes. When an NOE hydrogel was removed from the reaction solution, NO release did not go back to the initial baseline, suggesting some organoselenium species had diffused into the reaction solution. Quantitative analysis (Fig. 3e) uncovered that about $41 \pm 11\%$ of SeCA was still linked to the NOE hydrogels after catalytic generation of NO. The organoselenium species in the solution came from several sources, including remnant free SeCA in the NOE hydrogels, SeCA conjugated to uncross-linked A–M(9.3) molecules, and derivatives of SeCA as by-products (Fig. 3a). In the third scenario, any SeCA molecule linked to alginate with merely one amino group would lose half of its constituent part after catalysis. Finally, we measured the release rates of NO catalyzed by the NOE hydrogels after they had been preincubated in PBS for different durations. Our result shows that their catalytic potency could last for more than 2 weeks (Fig. 3f).

**Effects of the NOE hydrogels on cellular behaviors in vitro.** The integration of a vascular implant into the blood vessel is featured by the formation of neointima predominantly consisting of smooth

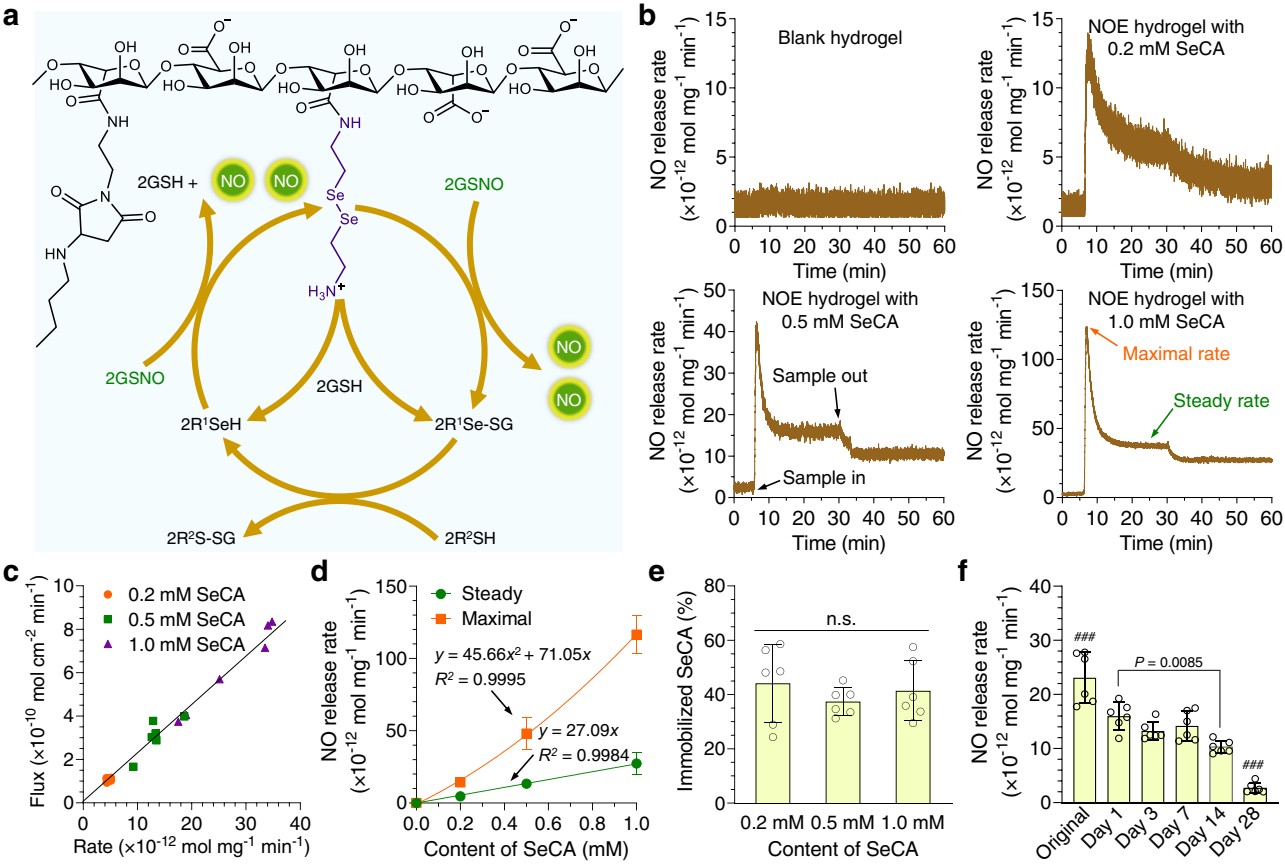

**Fig. 3 Catalytic generation of nitric oxide (NO) in the hydrogel. a** Mechanism on the catalytic generation of NO from NO donors (R'S-NO) by selenocystamine (SeCA) conjugated to alginate. **b** Representative curves of NO generation from S-nitrosoglutathione (GSNO, 10 μM in PBS at 37 °C) catalyzed by the nitric oxide-eluting (NOE) hydrogels conjugated with varying contents of SeCA (0.2–1.0 mM). **c** Correlation between the fluxes and rates of NO generation. **d** Summary on the release features of NO catalyzed by the NOE hydrogels (mean ± SD, $n = 6$ independent samples). **e** Quantification of the immobilized SeCA in the NOE hydrogels after catalytic generation of NO (mean ± SD, $n = 6$ independent samples). **f** Release rates of NO from GSNO catalyzed by the NOE hydrogel conjugated with 1.0 mM SeCA after preincubation in PBS at 37 °C for different durations (mean ± SD, $n = 6$ independent samples). One-way analysis of variance (ANOVA) with Tukey post hoc test was performed to determine the difference among various groups. (n.s. not significant, ###$P < 0.001$ compared with other groups).

muscle cells and/or endothelial cells. To prevent NIH, an ideal vascular stent must be capable of inhibiting vicinal smooth muscle cells while promoting rapid recruitment of endothelial cells. Previous studies have reported that NO at physiological level can inhibit smooth muscle cells while such effect does not act on endothelial cells[32,33]. To corroborate this, we started our investigation by conducting a competitive adhesion test between human umbilical vein endothelial cells (HUVECs) and human umbilical artery smooth muscle cells (HUASMCs) on our hydrogels in the medium supplemented with GSNO (10 μM) and glutathione (GSH, 30 μM). Our results (Figs. 4a and b) demonstrate that the number of HUASMCs adhering on the blank hydrogel within 3 h was less than a half of that on bare stainless steel. For the NOE hydrogels, the cell densities were even lower. In contrast, no significant difference in the density of adhering HUVECs was found among bare stainless steel and these hydrogels. Taken together, it can be concluded that the blank hydrogel selectively facilitates the adhesion of endothelial cells, while NO generation catalyzed by the NOE hydrogels further inhibits the attachment of smooth muscle cells.

To further evaluate the potential of our NOE hydrogels to promote endothelial regeneration, we seeded HUVECs onto them and cultured the cells in the presence of GSNO for prolonged time. Most of them attached onto the substrates within 6 h (Fig. 4c). Quantitative analyses (Figs. 4d–f) revealed no significant difference in the proliferation, coverage and spreading of

HUVECs among the blank hydrogel and NOE hydrogels, suggesting NO had no detrimental effects on their behaviors. In summary, the density and coverage of HUVECs on them increased from $147 \pm 20$ cells mm$^{-2}$ to $1010 \pm 61$ cells mm$^{-2}$, and $12.8 \pm 2.7\%$ to $96.9 \pm 3.0\%$, respectively, in 1 week, while the individual cell area almost unchanged. Compared with the hydrogels, the endothelial cells were much more spread on bare stainless steel even at early time. The individual cell area were $2126 \pm 385$ μm$^2$ after 6 h and $2295 \pm 243$ μm$^2$ after 3 days on it, while the values of these indices for the hydrogels were barely $863 \pm 97$ μm$^2$ and $990 \pm 101$ μm$^2$, respectively. However, no significant difference in cell-proliferation rate was noted among them. The direct consequence of better cell spreading on bare stainless steel was the higher cell coverage ($29.9 \pm 5.0\%$ after 6 h and $73.8 \pm 6.3\%$ after 3 days) in the early stage. Nonetheless, these indices became almost identical in one week among all groups.

It is worthy of note that HUVECs gathered as colonies and then propagated to form confluent monolayers on the hydrogels in 1 week, whereas those grown on bare stainless steel dispersed evenly and proliferated, until the formation of an intact cell sheet. Mauck et al.[34] have unraveled that cells are regulated by the interplay between cell–cell and cell–ECM interactions. On a stiff substrate like bare stainless steel in our case, the traction force sensed by HUVECs is relatively large, guiding them into a more spread phenotype through activating mechanotransduction

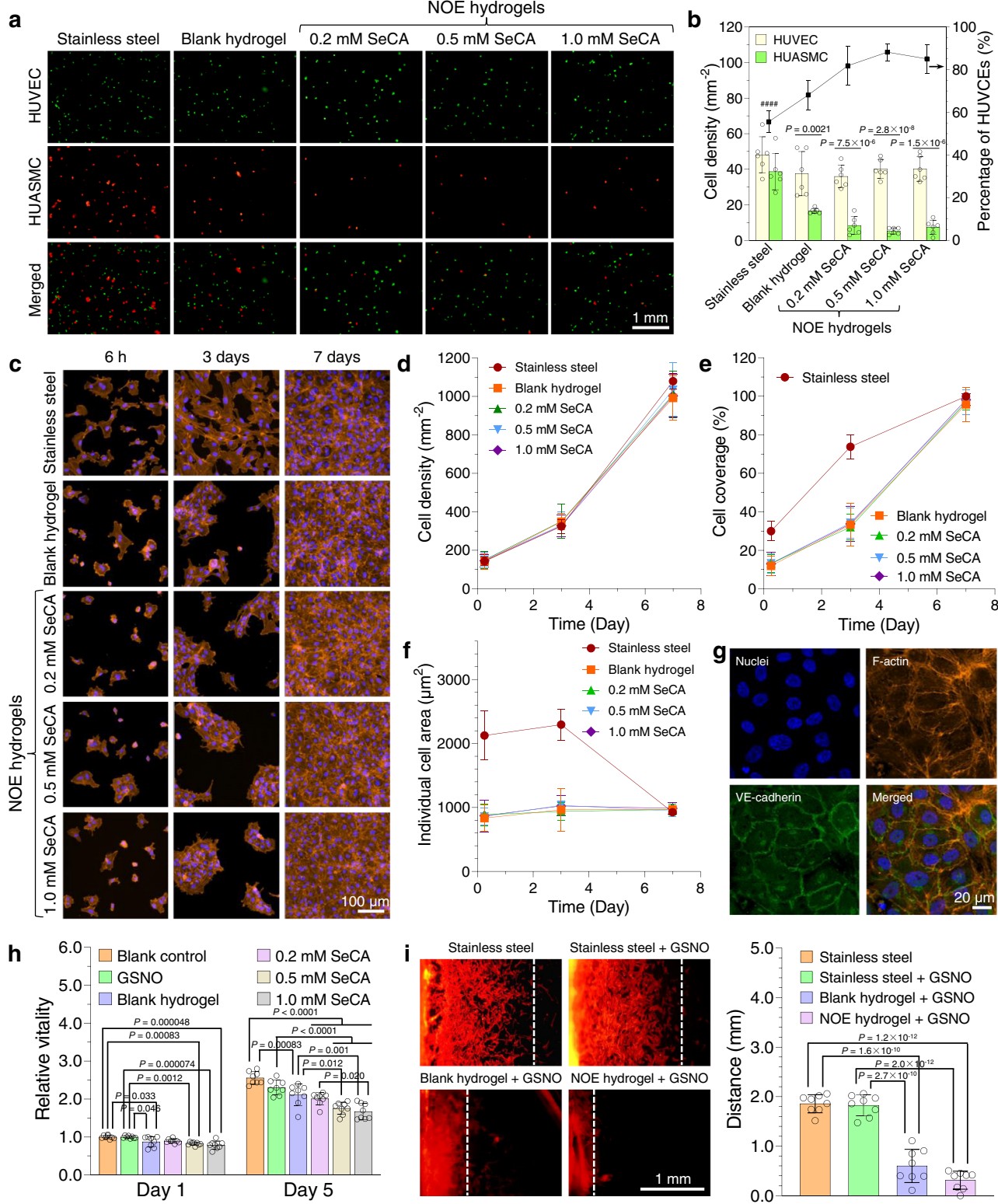

pathways such as YAP/TAZ[35]. On the contrary, those grown on our soft hydrogels were governed by cell–cell interaction due to the relatively low cell–ECM interaction, thereby leading to the formation of cell colonies and collective cell migration. At this stage, we could not conclude which scenario is more favorable for endothelial regeneration in vivo, but we observed that adherens junctions (VE-cadherin), which is necessary for healthy endothelium, had already formed between HUVECs grown on the hydrogels (Fig. 4g).

The competitive adhesion test implied that the hydrogels could inhibit smooth muscle cells. To verify this point of view, we cocultured HUASMCs with our hydrogels in the presence of GSNO. Cell-proliferation assay (Fig. 4h) suggested that GSNO alone had little influence on the cells, while their vitality was significantly reduced when cocultured with a hydrogel. The

**Fig. 4 Effects of the nitric oxide-eluting (NOE) hydrogels on cellular behaviors in vitro. a** Fluorescence images exhibiting the competitive adhesion between human umbilical vein endothelial cells (HUVECs) and human umbilical artery smooth muscle cells (HUASMCs) on various substrates. The cell growth medium was supplemented with S-nitrosoglutathione (GSNO, 10 µM) and glutathione (GSH, 30 µM). **b** Quantitative analyses on the competitive adhesion between HUVECs and HUASMCs (mean ± SD, $n = 6$ independent samples). **c** Confocal laser-scanning microscopy (CLSM) images displaying the adhesion, spreading and proliferation of HUVECs seeded onto various substrates in the presence of GSNO. **d–f**, Summary of cell density, cell coverage, and individual cell area for HUVECs on those substrates (mean ± SD, $n = 6$ independent samples). **g**, CLSM images showing the formation of adherens junctions (VE-cadherin) between HUVECs grown on the NOE hydrogel containing 1.0 mM selenocystamine (SeCA). Three independent samples were observed with similar results. **h**, Proliferation assay of HUASMCs cocultured with the blank hydrogel or NOE hydrogels containing varying contents of SeCA in the presence of GSNO (mean ± SD, $n = 8$ independent samples). **i**, Migration of HUASMCs on bare stainless steel, the blank hydrogel, and NOE hydrogel containing 1.0 mM SeCA (mean ± SD, $n = 8$ independent samples). One-way ANOVA with Tukey post hoc test was performed to determine the difference among various substrates and two-tailed Student's $t$-test was assumed to determine the difference between the two types of cells on the same substrate. ($^{####}P < 0.0001$ compared with other groups).

antiproliferative effect of the blank hydrogel might come from uncross-linked A–M(9.3) molecules in it, and the cell proliferation was further inhibited upon the generation of NO from the NOE hydrogels. Indeed, the cell vitality declined monotonically with the content of SeCA in them. In particular, after incubation with the NOE hydrogel containing 1.0 mM SeCA for 5 days, the vitality of HUASMCs decreased by 27% compared with those treated by GSNO alone. Due to the very limited amount of GSNO (the NO donor) dissolved in the small volume (1 mL) of cell medium, the generation of NO was unsustainable in vitro. Consequently, the NOE hydrogels were unable to inhibit the proliferation of smooth muscle cells at high efficiency. Nonetheless, this may not be an issue in vivo since the amount of GSNO in the blood of experimental animals could be two to three magnitudes higher.

We continued to evaluate the migration of HUASMCs on the hydrogels in 24 h according to a published protocol[36]. The experimental data (Fig. 4i) demonstrate that GSNO alone has little influence on the migration of HUASMCs since the distance traveled by them on bare stainless steel was unaffected by it (1.83 ± 0.16 mm with GSNO vs. 1.86 ± 0.17 mm without GSNO). To our delight, the movement of HUASMCs on the hydrogels was dramatically slowed down in comparison with those on bare stainless steel. The cells traveled 0.60 ± 0.33 mm on the blank hydrogel and 0.31 ± 0.17 mm on the NOE hydrogel containing 1.0 mM SeCA, respectively. These results indicate that the hydrogel material itself possesses some repressive effect on the migration of smooth muscles, while NO generation catalyzed by the NOE hydrogel could further retard this progress.

**Transcriptome analysis of HUASMCs cultured with the hydrogels**. To figure out why HUASMCs were inhibited by the blank hydrogel and NOE hydrogels, we performed a transcriptome analysis after the cells had been cocultured with them. The NOE hydrogel containing 1.0 mM SeCA was selected as the delegate since it was most efficient in inhibiting smooth muscle cells. Principal component analysis (PCA) and clustering assay (Fig. 5a and Supplementary Fig. 16) revealed that all three independent replicates in each group clustered together and GSNO alone barely had any impact on the gene expression of HUASMCs. However, with the coculture of the blank hydrogel plus GSNO, the phenotypic change of these cells became dramatic. For those incubated with the NOE hydrogel plus GSNO, such change was the largest. We set a threshold of $|\log_2$fold change (FC)$| > 1$ and $P < 0.05$ to screen out genes with significantly differential expression. Our results (Figs. 5b–d) disclose that 26, 244 and 565 genes presented significantly differential expression in HUASMCs after incubation with GSNO, the blank hydrogel plus GSNO, and the NOE hydrogel plus GSNO, respectively, for 6 h. Intriguingly, most of the alterations were

upregulated gene expression. Such data validate that GSNO alone has almost no influence on the phenotype of HUASMCs again, while the effects of the other two treatments are prominent. Compared with the blank hydrogel, the NOE hydrogel could catalyze the generation of NO from GSNO. The larger variation in gene expression of this group demonstrates that NO greatly affects the behavior of smooth muscle cells.

To better understand the effects of altered genomic expression profiles on cell behavior, these genes were analyzed in terms of inflammation, proliferation, and apoptosis. Although upregulation and downregulation were observed for both promotional genes and repressive genes, the general effects of these alterations were proinflammatory, antiproliferative and proapoptotic for the cells incubated with the blank hydrogel plus GSNO, while anti-inflammatory, anti-proliferative and proapoptotic for those incubated with the NOE hydrogel plus GSNO (Fig. 5e).

It is well known that NO affects smooth muscle cells through the canonical cGMP/PKG pathway[26,32,33] (Fig. 5f). It activates soluble guanylate cyclase (sGC), which subsequently catalyzes the transformation of guanosine triphosphate (GTP) into cyclic guanosine monophosphate (cGMP). cGMP can induce the relaxation of smooth muscle cells by interacting with cGMP-gated ion channels or play other biological functions through activating phosphate kinase G (PKG). In this study, we did observe the significant and unique upregulation of guanylate cyclase 1-soluble subunit alpha 2 (GUCY1A2) in HUASMCs after the treatment of NOE hydrogel plus GSNO. Besides, HMOX1[37] and PTGER4[38], which are two mediators for vascular relaxation, were also upregulated dramatically. These results confirm that our NOE hydrogel induced the relaxation of smooth muscle cells. Such phenomenon is highly desired because it can help the blood vessel to main vasodilation, thereby preventing the occlusion of stented artery. The mild inflammatory response in the group of blank hydrogel plus GSNO was likely to be elicited by gelatin since it was derived from animal tissues that might contain proinflammatory substances[39], yet the mechanism is not understood. In contrast, alginate might function as an anti-inflammatory mediator[40] so that the degree of inflammation was constrained. Encouragingly, NO molecules generated from the NOE hydrogel contributed extra anti-inflammatory modulation since a number of anti-inflammatory genes such as DUSP5[41] and NRG1[42] were exclusively upregulated (Fig. 5g). With respect to proliferation, NO was also found to trigger many unique antiproliferative alterations, such as the downregulation of proto-oncogenes CCND2[43] and SKP2[44], as well as the upregulation of tumor suppressor genes Nur77[45], PER2[46], GADD34[47], and FOXO1[48]. Most of these genes are involved in apoptosis at the same time. For instance, NUR77, PER2, and GADD34 are proapoptotic genes as well.

Summarizing the observations above, it can be concluded that the blank hydrogel induced mild inflammation in the smooth

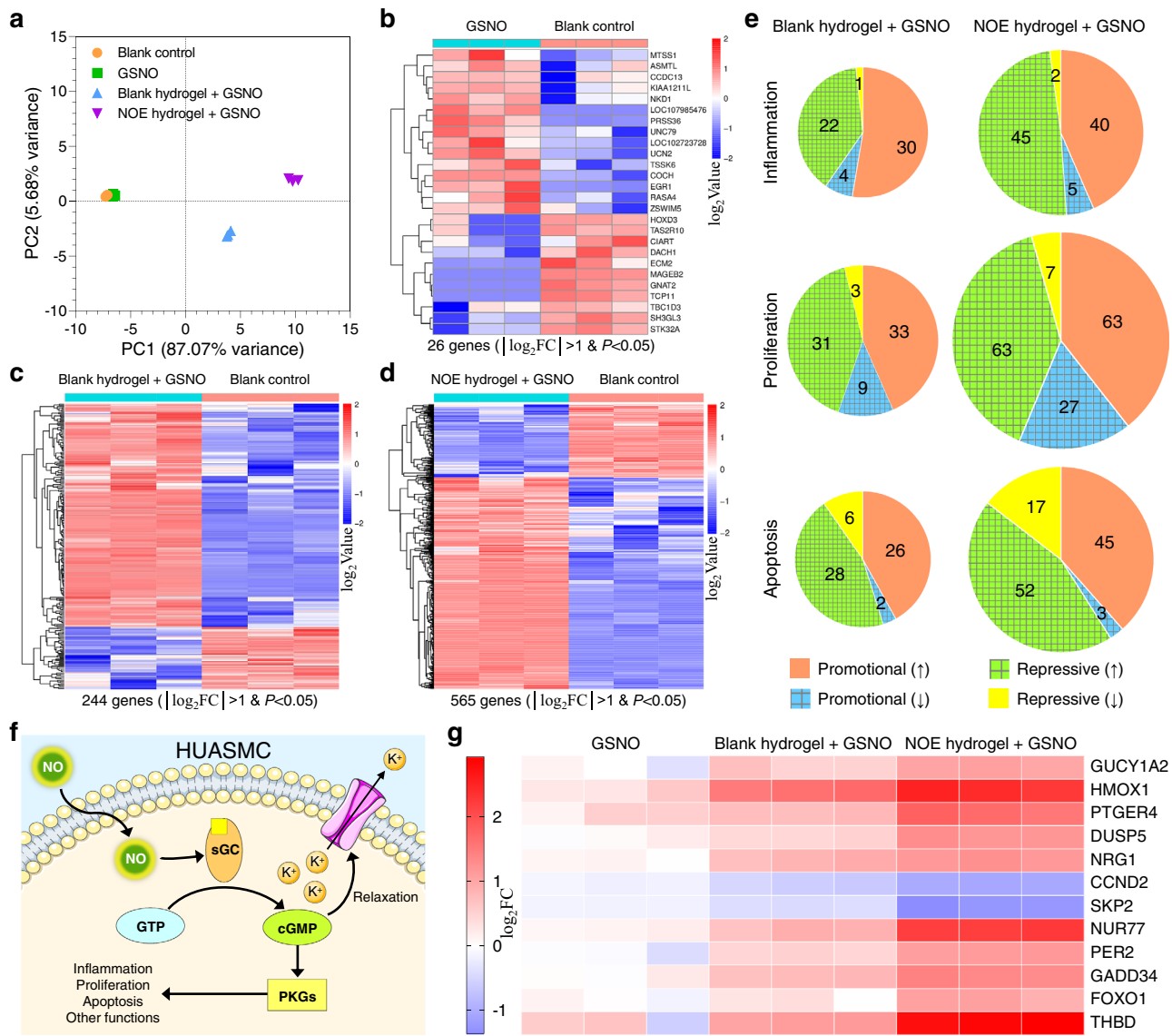

**Fig. 5 Transcriptome analysis of human umbilical artery smooth muscle cells (HUASMCs).** **a** Principal component analysis (PCA) representing the general variations in gene expression of HUASMCs among different groups. The nitric oxide-eluting (NOE) hydrogel containing 1.0 mM selenocystamine (SeCA) was selected as the delegate. **b**–**d** Differential gene expression heat maps of HUASMCs after various treatments when compared with the blank-control group. The gene expression levels for each set of comparison were normalized to the mean values within those two groups. Two-tailed Student's t-test was assumed to determine the difference between two groups. **e** Pie charts displaying the changes and numbers of significantly differential gene expression related to inflammation, proliferation, or apoptosis. The area of a pie represents the number of genes involved in. The gridded region represents the total number of anti-inflammatory, anti-proliferative, or antiapoptotic alterations in gene expression. **f** Schematic illustration of NO signaling pathway. **g** Heat map showing the relative changes in expression level for selected genes when compared with the blank-control group.

muscle cells even if it exerted antiproliferative and proapoptotic effects in the meanwhile. However, the catalytic generation of NO from the NOE hydrogel not only repressed such inflammation, but also intensified the antiproliferative and proapoptotic functions. These results corroborate the important role played by NO in the biological processes of smooth muscle cells.

**Vascular stent deployment in rabbit iliac arteries.** Encouraged by the results above, we continued to coat BMSs with our NOE hydrogel and test them in animals. Before that, the mechanical stability and thrombogenicity of the NOE hydrogel coating were examined. First, we conducted a mock angioplasty by dilating an NOE hydrogel-coated stent in a plastic catheter perfused with PBS (37 °C). Our results (Supplementary Fig. 17) demonstrate

that the NOE hydrogel was still intact even after being flushed by PBS for 1 week ($Q = 120$ mL min$^{-1}$ or $v = 28.3$ cm s$^{-1}$). Thereafter, we carried out a thrombogenicity test in an ex vivo arteriovenous shunt model (Supplementary Fig. 18 and 19). Our data show that both bare stainless steel and the blank hydrogel triggered severe clotting. In contrast, the NOE hydrogel could effectively retard blood coagulation or even completely inhibit it, depending on the content of conjugated SeCA (see Supplementary Information for detailed discussion). Indeed, the NOE hydrogel containing 1.0 mM SeCA could be regarded as non-thrombogenic. In view of its good biological performance, the NOE hydrogel containing 1.0 mM SeCA was selected as the coating material for BMSs.

We started our preclinical study by implanting the NOE hydrogel-coated stents (stent base: 316 L SS) into the right iliac

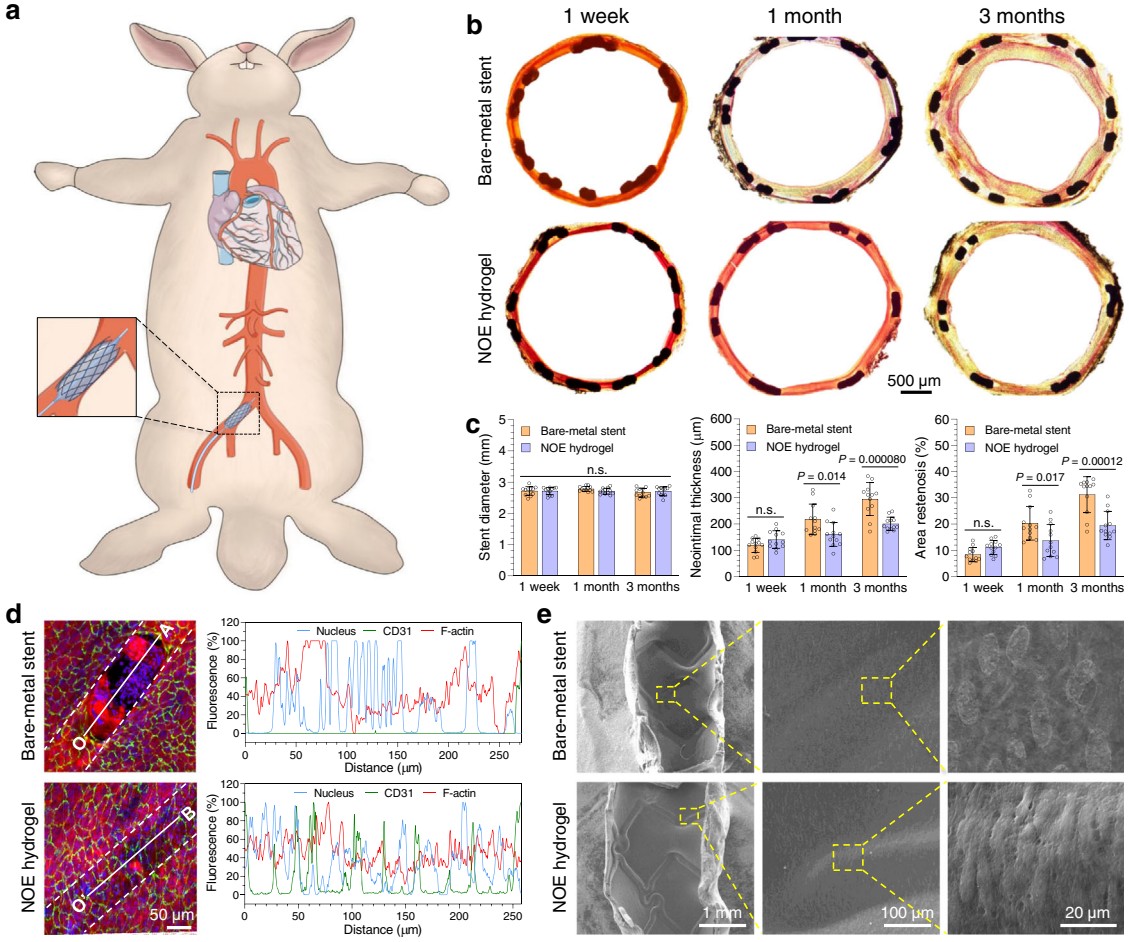

**Fig. 6 Vascular stent deployment in rabbits. a** Schematic illustration for vascular stent deployment in rabbit iliac arteries. **b** Optical images showing the cross sections of the stented arteries after van Gieson staining. **c** Quantitative analyses on the cross sections (mean ± SD, $n = 12$ independent animals). **d** Confocal laser-scanning microscopy (CLSM) unveiling the endothelialization on the stents (outlined by the dashed lines). The fluorescence intensities of different cell components along the line segments (OA, O'B) in the images were presented (blue: cell nucleus, green: CD31, red: F-actin). Six independent pairs of samples were observed with similar results. **e** Scanning electron microscopy (SEM) images showing the luminal faces of the stented arteries at 3 months post stent deployment. Three independent pairs of samples were observed with similar results. One-way ANOVA with Tukey post hoc test was performed to determine the difference among various groups and two-tailed Student's $t$-test was assumed to determine the difference between two groups. (*n.s.* not significant).

arteries of rabbits (Fig. 6a and Supplementary Fig. 20). BMSs of 316 L SS were used as the control and implanted into their left iliac arteries. At the designated time points, the stented arteries were harvested and examined. According to our observation, acute thrombosis (at 1 week post stent deployment) occurred on two BMSs, while NOE hydrogel-coated stent presented no thrombosis at all (Supplementary table 1). Van Gieson staining of their cross sections showed that all vascular stents were fully expanded, but neointimal growth varied dramatically among different groups (Fig. 6b). Quantitative analyses (Fig. 6c) suggested that the stent diameters were nearly identical and matched well with the reference value (2.7 mm) provided by the manufacturer of BMS. The neointimal thickness (NT) and area stenosis (AS) of them showed no difference within 1 week since implantation (Fig. 6c). However, neointima grew fast on BMS with NT increasing from $118 \pm 27$ μm to $295 \pm 63$ μm, and AS from $8.3 \pm 2.7\%$ to $31.2 \pm 6.9\%$ in 3 months (Fig. 6c and Supplementary Fig. 21). In contrast, these indices of NOE hydrogel-coated stent slowly increased to $201 \pm 25$ μm and $19.4 \pm 5.4\%$, respectively, which are significantly smaller ($P < 0.0001$) than those of BMS. Besides, the time-averaged neointimal growth rate on NOE hydrogel-coated stent decreased from 160 μm month$^{-1}$ in the first month to 20 μm month$^{-1}$ in the next two months, whereas these values for BMS are 217 μm month$^{-1}$ and 39 μm month$^{-1}$, respectively.

As aforementioned, our NOE hydrogel was expected to promote rapid restoration of native endothelium. To assess that, we utilized confocal laser-scanning microscope (CLSM) to examine the luminal faces of the stented arteries (Fig. 6d, Supplementary Video 2, Supplementary Video 3 and Supplementary Fig. 22). CLSM showed that NOE hydrogel-coated stent had already been covered with an intact endothelium in 1 week. In contrast, endothelization on BMS was incomplete during this period of time and some struts of it were not fully overlaid by endothelial cells, which was also corroborated by SEM (Supplementary Fig. 23). In addition, clusters of giant flat or small granular cells that seemed to be inflammatory cells were found on the nonendothelialized region. After implantation for 1 month, both types of stents were completely endothelialized (Supplementary Fig. 24). Nevertheless, the endothelial cells adhering on NOE hydrogel-coated stent presented a more mature phenotype compared with BMS, which was featured by elongated morphology and high degree of orientation (Supplementary Fig. 24 and Fig. 6e). Since our NOE hydrogel had presented anti-inflammatory effects in vitro, we also conducted histological

analyses on the cross sections of the stented arteries. Our results (Supplementary Fig. 25–28) demonstrate that BMS induced moderate to severe inflammation (see Supplementary Table 2 for the classification of inflammation) within 1 month, though the degree of inflammation reduced slightly after 3 months. In stark contrast, NOE hydrogel-coated stent only induced minimal inflammation during implantation.

In the cardiovascular system, the coordination between coagulation and fibrinolysis is critical for maintaining the intactness of blood vessels. During angioplasty, the vessel wall is injured inevitably, thereby releasing tissue factors that trigger coagulation. The ensuing formation of thrombus not only activates fibrinolysis, but also recruits inflammatory cells[49] as observed on BMS in our case. However, NO can suppress clotting cascade by preventing platelets from activation and may potentially inhibit thrombin through upregulation of thrombomodulin (THBD) in smooth muscle cells (Fig. 5g). Besides, the NOE hydrogel could provide a highly hydrated lubricating interface between the stent and blood, thereby reducing the turbulence of blood flow compared with BMS. It is reasonable to believe that thrombus formation was repressed on NOE hydrogel-coated stent as proved by the ex vivo thrombogenicity test. Consequently, the inflammation elicited by acute thrombosis was effectively prevented on NOE hydrogel-coated stent. In addition, the NOE hydrogel inhibited the proliferation of smooth muscle cells through the combinational effects of NO gas and A–M molecules. At the same time, it mimicked ECM, thereby providing a favorable microniche for endothelial cells. Thanks to these factors, NOE hydrogel-coated stent effectively suppressed NIH and presented faster restoration of native endothelium in comparison with BMS.

**Vascular stent deployment in porcine coronary arteries.** Although NOE hydrogel-coated stent demonstrated satisfactory outcomes in leporine model, those data might still be inadequate in predicting its performance in human since rabbits are herbivorous. Among large experimental animal species, the coronary artery system and physiology of pigs are very close to those of human, making them an ideal model for coronary stenting[50]. Consequently, we continued to evaluate NOE hydrogel-coated stent in a swine model. We compared it with an everolimus-eluting DES, which represents the gold standard for coronary stents. Both of them were constructed on the same type of cobalt chromium alloy (CoCr) stent, and the antirestenotic drug of DES was loaded in its polymer coating (see "Experimental" section for details). In addition, blank hydrogel-coated and blank polymer-coated stents were included as two negative controls (see "Experimental" section for details). These four types of stents were randomly implanted in three to four coronary arteries of individual *Bama* miniature pigs (Fig. 7a and Supplementary Fig. 29) under the guidance of digital subtraction angiography (DSA).

DSA was conducted again prior to the harvest of the stented arteries, which revealed all stented arteries were unobstructed in 2 weeks. However, severe narrowing occurred in the arteries implanted with polymer-coated stent (3 out of 6), while the blood flow was almost unaffected in other groups after 3 months (Fig. 7b; Supplementary Video 4 and 5). We harvested the stented arteries and examined them with naked eyes. Acute thrombosis (at 2 weeks post stent deployment) occurred on one polymer-coated stent and one blank hydrogel-coated stent, while other stents presented no thrombosis at all (Supplementary table 3). Our photographs (Fig. 7c) show that thick neointima grew on polymer-coated stent, while neointimal formation was much slower or even negligible on other stents. The cross sections of the

stented arteries were also stained (Fig. 7d). Quantitative analyses (Fig. 7e) revealed that the stent diameters were nearly identical among all groups. The NT value of polymer-coated stent was $439 \pm 71\,\mu m$ at 2 weeks and reached $596 \pm 123\,\mu m$ at 3 months post implantation, while AS increased from $50.9 \pm 8.0\%$ to $66.8 \pm 10.7\%$ during that time. Moreover, all the three polymer-coated stents displaying severe narrowing under DSA presented in-stent restenosis (diameter stenosis >50%). Upon the loading of everolimus in the polymer coating, DES strongly inhibited neointimal growth. The NT and AS for DES were barely $124 \pm 18\,\mu m$ and $13.2 \pm 2.8\%$, respectively, after 2 weeks. However, such inhibitory effect was unsustainable as we found that these indices increased to $327 \pm 49\,\mu m$ and $36.4 \pm 5.4\%$ in 3 months. In contrast, NOE hydrogel-coated stent presented both efficient and sustained suppression of NIH. Although this type of stent was slightly inferior in the short term, it defeated DES in the long run since the NT and AS increased merely to $206 \pm 29\,\mu m$ and $23.0 \pm 3.1\%$, respectively, in 3 months. In addition, the NIH on blank hydrogel-coated stent was significantly less severe than that of polymer-coated one. As a matter of fact, the NT and AS of blank hydrogel-coated stent were even close to the level of DES in 3 months.

To explain the difference among various stents, we also examined the status of endothelialization on them. CLSM and SEM images (Fig. 7f and g; Supplementary Fig. 30 and 31) disclose that endothelialization was dramatically delayed on DES compared with other stents. In fact, the progress of endothelialization on it was still incomplete even after 3 months. In stark contrast, both NOE hydrogel-coated and blank hydrogel-coated stents promoted rapid restoration of endothelium in 2 weeks. However, the endothelial cells on blank hydrogel-coated stent displayed loose contact with each other and were less oriented compared with those on NOE hydrogel-coated one. Besides, the strut profile was invisible for blank hydrogel-coated stent due to the thick neointima. When it comes to polymer-coated stent, though high degree of endothelialization was observed in 2 weeks, the endothelial cells were highly elongated and oriented in the direction of blood flow. In addition, the newly formed endothelium was dispersed with holes. At 3 months post implantation, the endothelium on polymer-coated stent became intact, but the endothelial cells were further stretched due to the large shear stress of blood flow caused by NIH. We plotted the increments in neointimal thickness between 2 weeks and 3 months post stent deployment versus the mean endothelial coverage during that period of time (Fig. 7h). Our data suggest that NIH is negatively correlated with the degree of endothelialization within this duration. In particular, neointimal growth was strongly suppressed on NOE hydrogel-coated stent, for which an intact endothelium had formed in 2 weeks. Histological analysis (Supplementary Fig. 32 to 34a) revealed that polymer-coated stent induced severe inflammation during implantation, while mild-to-moderate inflammation was observed on other stents. DES effectively repressed inflammation within 2 weeks, but such repressive effect was unsustainable because the inflammation in the stented arteries exacerbated after 3 months. In contrast, NOE hydrogel-coated stent persistently suppressed inflammation. Notably, although blank hydrogel-coated stent induced moderate inflammation in the beginning, such inflammation mitigated after 3 months.

Based on the observations above, we can rationally conclude that polymer-coated stent is highly proinflammatory, thereby stimulating NIH in the stented artery (Supplementary Fig. 34b). In comparison with it, blank hydrogel-coated stent is more biocompatible, so that the neointimal formation was much slower on it. In the case of DES, everolimus released from the polymer coating only temporarily repressed inflammation and neointimal

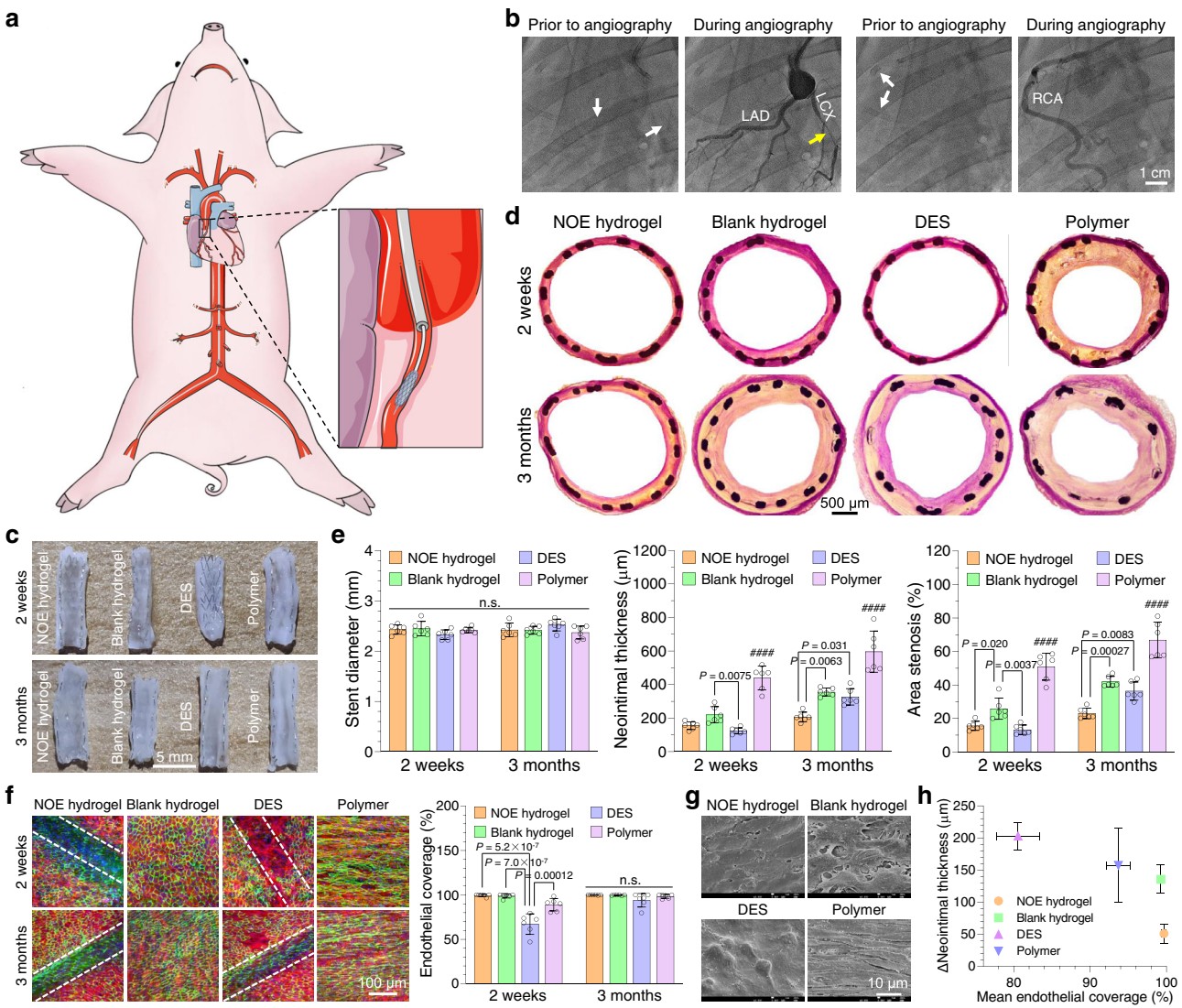

**Fig. 7 Vascular stent deployment in pigs. a** Schematic illustration for vascular stent deployment in porcine coronary arteries. **b** Digital subtraction angiography prior to the harvest of the stented arteries. The white arrows indicate the sites of implanted stents. The yellow arrow refers to severe restenosis occurring in a polymer-coated stent. **c** Photographs displaying the luminal faces of the stented coronary arteries at 2 weeks and 3 months post stent deployment. **d** Optical images showing the cross sections of the stented arteries after van Gieson staining. **e** Quantitative analyses on the cross-sections (mean ± SD, $n = 6$ independent animals). **f** Confocal laser-scanning microscopy (CLSM) unveiling the endothelialization on the stents (outlined by the dashed lines). (blue: cell nucleus, green: CD31, red: F-actin). The endothelial coverages were determined for different types of stents (mean ± SD, $n = 6$ independent animals). **g** Scanning electron microscopy (SEM) images showing the luminal faces of the stented arteries at 2 weeks and 3 months post stent deployment. **h** Correlation between endothelialization and neointimal hyperplasia. The increments in neointimal thickness between 2 weeks and 3 months post stent deployment were plotted versus the mean endothelial coverage during that period of time (mean ± SEM, $n = 6$ independent animals). One-way ANOVA with Tukey post hoc test was performed to determine the difference among various groups ($^{####}P < 0.0001$ compared with other groups).

growth. Besides, the anti-restenotic drug impaired the regeneration of native endothelium. Consequently, inflammatory response was still elicited upon the depletion of the drug, causing NIH on DES at late stage. In contrast, our NOE hydrogel not only repressed inflammation, but also promoted rapid restoration of native endothelium. As a result, NOE hydrogel-coated stent persistently suppressed NIH.

To present the advancement of our NOE hydrogel more convincingly, we did a literature review and compared NOE hydrogel-coated stent with other stents deployed in the iliac arteries of healthy, balloon-injured, or high-fat diet fed rabbits. Our summary (Supplementary Table 4) reflects that conventional DESs are generally potent in suppressing NIH. However, the progress of endothelialization on them is markedly delayed, and it is still

incomplete after 3 months on some DESs. BMSs are favorable for the restoration of endothelium, but they normally possess high thrombogenicity and induce thicker neointimal formation. The FDA-approved fully bioresorbable stent Absorb BVS® fails in all terms of thrombogenicity, endothelialization and NIH. Other stents are thrombogenic, inefficient in promoting endothelialization, or incompetent in suppressing NIH. In stark contrast, NOE hydrogel-coated stent is nonthrombogenic and achieves complete endothelial regeneration in 1 week. In the meantime, it is comparable with DESs in view of anti-restenosis. Taking thrombogenicity, endothelialization and NIH into consideration together, NOE hydrogel-coated stent is best in overall performance.

## Discussion

The emergence of vascular stents has saved millions of patients with coronary artery disease. However, in-stent restenosis, as a result of neointimal hyperplasia or stent thrombosis, is a great challenge for the approved stents. The nitric oxide-eluting (NOE) hydrogel developed by us shed a light on the resolution of this issue. It is mechanically tough and could provide sustained generation of nitric oxide to inhibit thrombosis and repress inflammation. Besides, it could promote rapid endothelial regeneration thanks to its chemical resemblance to the extracellular matrix. Because of these synergistic effects, the NOE hydrogel coating persistently suppressed neointimal hyperplasia on vascular stent.

However, it should be mentioned that some limitations also exist in the design of our animal experiment. The drug-eluting stent (DES) assumed in swine model is customized based on a cobalt chromium alloy stent, so that its performance may not accurately reflect those of approved products. For instance, the neointimal hyperplasia on it became severe at 3 months post implantation, which might have been resolved in commercial DESs by loading more anti-restenotic drug or using a more compact polymer coating. In addition, our study was conducted on healthy animals, whereas the biological effects of NO on pathological models are not clear. A thorough understanding of how NO acts on pathological cardiovascular system is still needed to ensure its efficacy and safety for clinical applications[51].

Nevertheless, these limitations cannot overshadow the brilliance of our NOE hydrogel. Apart from vascular stent, it may be assumed as the coating material for other vascular implants, including artificial valves and blood vessels. Besides, this hydrogel is self-cross-linking and can be conjugated with various bioactive molecules due to its abundant functional groups. Moreover, it allows the carrying of small organic or large protein-based medicine (Supplementary Fig. 35) to afford extra therapy. Consequently, we expect that such hydrogel will find wide applications in biomedical engineering, such as scaffolds for tissue engineering and dressings for wound healing.

## Methods

**Materials.** Alginate (MW: 10~20 kDa) and gelatin (Type A, Cat. No.: G1890) were purchased from Qingdao Hyzlin Biology Development Co., Ltd. (China) and Sigma-Aldrich (USA), respectively. N-(2-aminoethyl)maleimidetrifluoroacetate salt (AEM.TFA), N-hydroxysulfosuccinimide (Sulfo-NHS), and 1-ethyl-3-(3-dimethyl-laminopropyl)carbodiimide hydrochloride (EDC.HCl) were supplied by Suzhou Highfine Biotech (China). Glycinamide hydrochloride (Cat. No.: 459977) and methacrylic anhydride (Cat. No.: L14357) were obtained from J&K Scientific (China) and Alfa Aesar (USA), respectively. Selenocystamine dihydrochloride (SeCA.2HCl, Cat. No.: S0520), S-nitrosoglutathione (GSNO, Cat. No.: N4148), and L-glutathione (reduced, GSH, Cat. No.: G4251) were bought from Sigma-Aldrich. Polydimethylsiloxane (PDMS) prepolymer (SYLGARD® 184) was provided by Dow Corning (USA). Pentobarbital sodium salt (Cat. No.: P0776), benzylpenicillin sodium salt (Cat. No.: 150061248) and heparin (Cat. No.: BP252410) were acquired from TCI (Japan), Shandong Shengwang Pharmaceutical Co., Ltd. (China), and Thermo Fisher Scientific (USA), respectively. Other chemicals were purchased from Sigma-Aldrich and used without further purification unless stated otherwise.

**Syntheses of A–M.** To synthesize A–M, pristine alginate (4.2 g) was dissolved in PBS (100 mL) at 50 °C and then centrifuged at 6,000 g for 15 min to deposit insoluble impurities. Subsequently, the supernatant was pushed through 0.45 μm and 0.22 μm filters (Minisart®, Cat. No.: 16555 and 16532, Sartorius, Germany) consecutively, and collected in a clean glass bottle (250 mL). For the coupling of maleimide to alginate, AEM.TFA (508.4 mg, 1016.8 mg, or 1525.2 mg; 2 mmol, 4 mmol, or 6 mmol), Sulfo-NHS (521.2 mg, 1042.4 mg, or 1563.6 mg; 2.4 mmol, 4.8 mmol or 7.2 mmol), and EDC.HCl (1.38 g, 2.76 g, or 4.14 g; 7.2 mmol, 14.4 mmol or 21.6 mmol) were added into the solution and mixed by magnetic stirring. To obtain A–M with the potency to catalyze NO generation, the corresponding solution was also supplemented with SeCA (42.4 mg; 0.133 mmol). The reaction was carried out at ambient temperature in the dark for the designated period of time (6 h, 8 h or 10 h). Afterward, the pH value of the solution was adjusted to about 4.5 by HCl solution (1 M, 3 mL), and then the reaction mixture was dialyzed (MWCO: 12–14 kDa, Spectra/Por 4, Spectrum Laboratories Inc., USA) against diluted HCl solution (pH 4.5) for 2 days. Finally, the solution was pushed through a 0.22-μm filter, flash-frozen in liquid nitrogen, lyophilized and then stored in a −80 °C freezer (~80% yield).

**Syntheses of GelGA and GelMA.** To synthesize GelGA, pristine gelatin (2.56 g) was dissolved in PBS (40 mL) at 45 °C. Subsequently, glycinamide hydrochloride (4.42 g; 40 mmol), Sulfo-NHS (0.87 g; 4 mmol), and EDC.HCl (2.30 g; 12 mmol) were added into the solution and mixed by magnetic stirring. The reaction was carried out at 45 °C in the dark for 5 h after the pH value of the solution had been adjusted to about 7.5 by NaOH solution (1 M, 8 mL). After that, the biopolymer was precipitated in ethanol (500 mL). The precipitant was collected by filtration, redissolved in hot water (40 mL), and then dialyzed against warm water (35 °C) for 2 days. Finally, the purified solution was pushed through a 0.22-μm filter, flash-frozen in liquid nitrogen, lyophilized and then stored in a −20 °C freezer (~70% yield).

The synthesis of GelMA was conducted according to the published protocol[52]. Briefly, pristine gelatin (5 g) was dissolved in PBS (50 mL) at 50 °C. Subsequently, methacrylate anhydride (4 mL) was added dropwise to the gelatin solution while being stirred. After 6 h, the solution was dialyzed against warm water (35 °C) for 5 days to remove unreacted monomers and salts. The purified solution was also pushed through a 0.22 μm filter, flash-frozen in liquid nitrogen, lyophilized, and then stored in the −20 °C freezer (~70% yield).

**$^1$H-NMR spectroscopy for the biopolymers.** To determine the $^1$H-NMR spectra of A–M, GelGA, and GelMA, the biopolymers were dissolved in $D_2O$ at 40 °C to the mass concentration of around 10 mg mL$^{-1}$. Afterward, the freshly prepared specimens were analyzed by a 400-MHz NMR spectrometer (Avance II, Bruker; software: Bruker TopSpin 3.5.6), with scan times being at least 32 to achieve high signal-to-noise (SNR) ratios. The spectra were calibrated by setting the chemical shift (δ) of HOD to 4.8 ppm. The generated data were analyzed with MestReNova 14.0.0 (Mestrelab Research, Spain).

**Preparation of the hydrogels.** The precursor solutions of the biopolymers were prepared by dissolving them in phosphate buffer (PB, 10 mM, pH 7.4) to the mass concentration of 150 mg mL$^{-1}$ or 100 mg mL$^{-1}$ with final pH adjusted to about 7.5 using NaOH solution (1 M). To fabricate the hydrogels of A–M/G, the corresponding precursor solutions (at 60 °C) were mixed at the designated mass ratios using a pipette gun and then cured in a humidified atmosphere at 37 °C and under 5% $CO_2$. The pristine gelatin, GelGA and GelMA hydrogels were made by simply cooling down the precursor solutions (at 60 °C) in a humidified atmosphere at ambient conditions. The photo-cross-linked GelMA hydrogels were produced by further illuminating the physically cross-linked GelMA hydrogels (containing 0.3 wt% Irgacure 2959) with UV light (365 nm, 8 W, ENF-280C, Spectronics Corporation, USA) for 30 min.

**Dynamic mechanical analysis of the hydrogels.** Hydrogels in disc shapes (diameter: 8 mm, thickness: 1 mm) were prepared in a homemade well plate formed by bonding a perforated plastic plate to a piece of polydimethylsiloxane (PDMS) membrane. At the designated time points, the hydrogel disc was carefully demolded and transferred to a parallel plate fixture (8 mm in diameter) mounted on a rheometer (ARES, TA Instruments, USA; software: TA Orchestrator 7.2.0.4). The upper plate was slowly moved down until close contact with the hydrogel. For strain sweep (1–100%), dynamic mechanical test was carried out at 1 rad s$^{-1}$ angular frequency. The cure kinetics for the hydrogels was investigated in time-sweep mode at 5% shear strain and 1 rad s$^{-1}$ angular frequency.

**Tensile experiment of the hydrogels.** To prepare hydrogels for tensile experiment, the precursor solutions were injected into homemade dumbbell-shape PDMS molds (middle part: 20 mm × 5 mm, end parts: 5 mm × 9 mm; depth: 2 mm). After cure, the hydrogel was carefully demolded and transferred onto a thin-film fixture mounted on the rheometer. Two homemade PDMS clamps with the complementary shape to the ends of the hydrogel were used to fix it to the fixture. Thereafter, the tensile test was conducted on the hydrogel by stretching it at the rate of 0.1 mm s$^{-1}$ until fracture.

**CD spectroscopy for the gelatin-based hydrogels.** Pristine gelatin, GelGA, or GelMA solution (150 mg mL$^{-1}$) was cured in the well (path length: 0.1 mm, volume: 16 μL) of a quartz plate and then examined by a spectropolarimeter (Chirascan V100, Applied Photophysics, UK; software: Pro-Data Chirascan 4.4.2.0) at ambient temperature. The CD spectra (range: 200–320 nm, interval: 1 nm) of the gelatin-based hydrogels were generated from the averaged value of three scans (20 nm min$^{-1}$), which were smoothened by the software provided by the manufacturer of the equipment.

**Melting points of the gelatin-based hydrogels.** To determine the melting points of the gelatin-based hydrogels (150 mg mL$^{-1}$), dynamic mechanical test was performed in temperature sweep mode (23–40 °C) at 5% shear strain and 1 rad s$^{-1}$ angular frequency. The ramp of temperature was set as 2 °C min$^{-1}$ and the sampling frequency was between 0.008 Hz and 0.016 Hz. We defined the melting

point of a hydrogel as the temperature at which the shear modulus was reduced by a half from its initial value.

**Culture of HUVECs and HUASMCs.** Human umbilical vein endothelial cells (HUVECs, Cat. No.: C-12203) and human umbilical artery smooth muscle cells (HUASMCs, Cat. No.: C-12500) were purchased from Promocell (USA) and cultured in endothelial cell growth medium (Cat. No.: C-22010, Promocell) or smooth muscle cell growth medium (Cat. No.: C-22062, Promocell), respectively. Both media were supplemented with 1 v/v% penicillin/streptomycin solution (Cat. No.: 15140122, Thermo Fisher Scientific) and 1 v/v% amphotericin B solution (Cat. No.: LS15290026, Thermo Fisher Scientific). The cells were raised at 37 °C and under 5% $CO_2$ on tissue culture dishes (Cat. No.: 93100, TPP, Switzerland), until they reached about 80% confluence. To detach the cells, they were rinsed with PBS and then treated with trypsin (0.05 w/v%)/EDTA (0.53 mM) solution (Cat. No.: 15400054, Thermo Fisher Scientific). The floated cells were deposited after centrifugation and then resuspended in their own growth medium for subculture or biological tests.

**Catalytic generation of NO from the NOE hydrogels.** The NO-generating capacity of an NOE hydrogel was evaluated by measuring the release rate of NO catalyzed by it using a highly sensitive and selective analyzer (Sievers NOA 280i, GE Healthcare, USA; software: NOAnalysis 3.2). Briefly, 316 L stainless-steel (SS) foils (10 mm × 5 mm × 0.05 mm) were predeposited with P(DA-co-HDA)[23] and then covered with a precursor solution (~10 µL) of the NOE hydrogel. After full cure, the specimen was punched, hanged onto a stainless-steel wire, and then plunged into PBS solution (5 mL) containing 10 µM GSNO and 30 µM GSH at 37 °C. The reaction solution was purged with $N_2$ gas and the generated NO was carried to the reaction chamber of the analyzer, wherein NO was oxidized by $O_3$ into $NO_2$ at its excited state ($NO_2^*$). Upon the relaxation of $NO_2^*$, a photon was emitted, which could be detected by the analyzer for the quantification of NO.

**Sterilization of the specimens.** For in vitro and in vivo studies, the as-prepared specimens were sterilized with 75 v/v% alcohol. Subsequently, they were rinsed with 50 v/v%, 25 v/v% and 10 v/v% alcohols in sequence. Afterward, they were bathed with PBS for immediate use, or washed with pure water and then dried for storage.

**Competitive adhesion between HUVECs and HUASMCs.** HUVECs and HUASMCs were prelabeled with CellTracker™ Green CMFDA (Cat. No.: C2925, Thermo Fisher Scientific) and Orange CMTMR (Cat. No.: C2927, Thermo Fisher Scientific), respectively, according to the protocols provided by the vendor. Subsequently, the cells were detached, deposited, and then resuspeneded in their own growth medium supplemented with GSNO (10 µM) and GSH (30 µM). The two types of cells were diluted to the same density of 5000 cells $mL^{-1}$, mixed at the ratio of 1:1, and then seeded onto the substrates placed in the 12-well plates at the density of 10,000 cells $cm^{-2}$. After incubation for 3 h, the cells were gently washed with PBS and then photographed using an inverted fluorescence microscope (CKX53, Olympus, Japan; software: Olympus CellSens 2.3).

**Proliferation of HUASMCs cocultured with the hydrogel.** HUASMCs were seeded in 24-well plates (Cat. No.: 92024, TPP) at the density of $1 \times 10^4$ cells $cm^{-2}$. After incubation for overnight, they were cocultured with the blank hydrogel or NOE hydrogel (disc with a diameter of 6.4 mm, ~12 mg) in the medium (0.4 mL) supplemented with GSNO (10 mM) and GSH (30 mM). The medium was refreshed every day with the replenishment of GSNO and GSH at 1 h, 2 h, 3 h, 6 h, 12 h and then every 12 h since coculture. At day 1 and day 5 since coculture, the vitalities of the cells were measured using Cell Counting Kit-8 (CCK-8, Dojindo Molecular Technologies, Japan) following the protocol provided by the vendor and normalized to the value of blank control at day 1. As another control, the vitality of HUASMCs raised in the medium containing GSNO and GSH was also determined.

**Migration test of HUASMCs on the substrates.** HUASMCs were seeded onto L-shape-folded stainless-steel foils (2 cm × 1 cm) coated by the blank hydrogel or NOE hydrogel (1.0 mM SeCA) at the density of $5 \times 10^4$ cells $cm^{-2}$ in 24-well plates. After incubation for overnight, the unseeded half-parts were flipped down and submerged in the medium (2 mL) supplemented with GSNO and GSH. The cells were allowed to migrate freely for 24 h, during which GSNO and GSH were replenished at 1 h, 2 h, 3 h, 6 h, and 12 h since the test. Subsequently, the specimens were fixed with paraformaldehyde (4 w/v% in PBS) for 1 h, permeabilized with Triton X-100 solution (0.2 v/v% in PBS) for 1 h, blocked with BSA solution (5 w/v % in PBS) for 12 h, and stained with phalloidin-TRITC (1 µg $mL^{-1}$ in PBS) for 3 h sequentially. After washing with PBS for three times and cutting off the cell-seeded half-parts of the foils, the cells were imaged using a fluorescence microscope (IX51, Olympus, Japan; software: Olympus CellSens 2.3). As controls, the migration of HUASMCs on bare stainless-steel foils in the presence or absence of GSNO and GSH was tested as well.

**Adhesion and growth of HUVECs on the substrates.** The blank hydrogel or NOE hydrogels (~45 µL) were formed on mirror polished stainless-steel sheets (15 mm × 15 mm × 0.5 mm) predeposited with P(DA-co-HDA)[23]. The as-prepared specimens were placed into 12-well plates (Cat. No.: 92012, TPP) and then plated with HUVECs at the density of 20,000 cells $cm^{-2}$. The cells were cultured in the medium supplemented with GSNO (10 µM) and GSH (30 µM) for different periods of time, during which the medium was renewed every day. At the designated time, they were fixed with paraformaldehyde solution for 1 h, permeabilized with Triton X-100 solution for 1 h and then blocked with BSA solution for 12 h. Subsequently, they were incubated with Alexa Fluor® 488-conjugated rabbit anti-human VE cadherin monoclonal antibody (Clone No.: EPR18229, Cat. No.: ab225443, Abcam, USA, 1/100 dilution in PBS) for 6 h, followed by staining with phalloidin-TRITC (1 µg $mL^{-1}$ in PBS) and DAPI (5 µg $mL^{-1}$ in PBS) for 6 h. After extensive washing with PBS, the cells on the substrates were photographed using a confocal laser-scanning microscope (CLSM, TCS SP5 II, Leica, Germany; software: Leica Application Suite X).

**Transcriptome analysis of HUASMCs.** HUASMCs were seeded in tissue culture dishes (Cat. No.: 93100, TPP) at the density of $2 \times 10^4$ cells $cm^{-2}$. After incubation for overnight, they were cocultured with the blank hydrogel or NOE hydrogel (disc with a diameter of 3.5 cm, ~400 mg) in the medium (10 mL) supplemented with GSNO (10 mM) and GSH (30 mM). The medium was replenished with GSNO and GSH at 1 h, 2 h, and 3 h since the coculture. After incubation for 6 h, both the medium and coatings were removed. Subsequently, the total RNA of the cells was extracted by TRIzol® reagent (Thermo Fisher Scientific, USA) following the protocol provided by the vendor. As controls, the total RNA was also extracted for the cells at normal culture condition or incubated with GSNO and GSH. The transcriptome sequencing and analysis were conducted by OE biotech Co., Ltd. (Shanghai, China). Raw data were processed using Trimmomatic. The reads containing poly-N and low quality reads were removed to obtain clean reads, which were mapped to reference genome using HISAT2. FPKM value of each gene was calculated using Cufflinks, and the read count of each gene was obtained by HTSeq-Count. Differentially expressed genes (DEGs) were identified using DESeq R package 1.18.0 functions estimateSizeFactors and nbiomTest. The threshold of $|\log_2 FC| > 1$ and $P < 0.05$ was set for significantly differential expression. Hierarchical cluster analysis of DEGs was performed to explore transcript expression patterns.

**Preparation of the hydrogel coating on vascular stent.** Bare BMSs (18 mm × 2.70 mm; 316 L SS for leporine model and CrCo for swine model; Kossel Medtech Co., Ltd., China) were used as the stent base and/or control in this study. For the 316 L SS stent, the width and thickness of the struts are 97 µm and 100 µm, respectively. The strut-to-artery surface area ratio of it is 0.153. For the CrCo stent, the width and thickness of the struts are 85 µm and 89 µm, respectively. The strut-to-artery surface area ratio of it is 0.147. The BMSs were predeposited with P(DA-co-HDA)[22] and then dipped into the precursor solution of the blank hydrogel or NOE hydrogel. The stents were carefully pulled out with tweezers and rolled on clean glass slides to remove excess liquid. Subsequently, the precursor solution left on the stents was cured at 37 °C in a humidified atmosphere for 12 h. The dip-coating procedure was repeated twice to increase the thickness of the hydrogel coating. Afterward, the hydrogel coating was allowed to cure for another 2 days.

**Mechanical stability testing of the hydrogel coatings.** Two BMSs of 316 L SS were predeposited with P(DA-co-HDA)[23] and then dip-coated with the blank hydrogel or NOE hydrogel. Afterward, they were loaded onto balloon catheters and compressed until close contact. The stent coated with the blank hydrogel was dilated in PBS at 37 °C by inflating the balloon, until the pressure reached 8 MPa, which was maintained for one minute. The other stent coated with the NOE hydrogel was dilated in a catheter ($D_{in}$: 3 mm, $D_{out}$: 4 mm) and then constantly flushed by PBS (37 °C, $Q = 120$ mL $min^{-1}$ or $v = 28.3$ cm $s^{-1}$) for 1 week. Both stents were photographed using the inverted fluorescence microscope (CKX53) before and after testing.

**Thrombogenicity test in extracorporeal circulation.** A total of 8 male *New Zealand* white rabbits (~4 months old, 2.5 ~ 3.0 kg) were used in this test. Each rabbit was anesthetized by intravenous injection (through marginal ear vein) of pentobarbital sodium salt solution (30 mg $mL^{-1}$ in saline, 1 mL $kg^{-1}$). The left carotid artery and right external jugular vein of the rabbit were isolated immediately. Subsequently, the animal was intravenously injected with GSNO (10 mM in saline, 0.1 mL $kg^{-1}$) and GSH (30 mM in saline, 0.1 mL $kg^{-1}$) solutions. Afterward, an arteriovenous extracorporeal circuit was established by cannulating the left carotid artery and the right jugular vein. The circuit had four parallel shunt catheters in the middle and each contained a curled 316 L SS foil (15 mm × 8 mm × 0.02 mm) that was either bare or coated with the hydrogel (~24 µL). The blood flow through the circuit was started by removing the hemostatic clamps on the artery and vein. After 3 h, the blood flow was stopped using the clamps again, and the animal was euthanized by injecting excess pentobarbital sodium salt solution (3~5 mL) when the circuit had been removed. The catheter segments containing the specimens were cut with scissors, extensively flushed by heparin

solution (50 U mL$^{-1}$) and then photographed. The thrombogenicity of a specimen was assessed by measuring the occlusion and patency of the host catheter segment as well as the weight of thrombus formed on the guest foil. After that, the specimens were fixed by glutaraldehyde (2.5 v/v% in saline), dehydrated by gradient alcohols (50 v/v%, 75 v/v%, 90 v/v%, and 100 v/v%, each for 1 h), dried in air, sputter-coated with gold, and then examined using a scanning electron microscope (SEM, Quanta 200, FEI, USA; software: xT Microscope Control 2.2).

**Vascular stent deployment in rabbits**. A total of 36 male *New Zealand* white rabbits (~3 months old, 2.0~2.5 kg) were used for this study, which were raised with high-fat diet. The vascular stent coated with the NOE hydrogel containing 1.0 mM SeCA was assumed as the experimental group and BMS of 316 L SS as the control group. Two stents (one for each group) were implanted bilaterally into the left (experimental group) and right (control group) iliac arteries of each rabbit. The arteries were overdilated to induce injury by increasing the pressure of the balloon catheter to 10 atm and maintaining it for 1 min. The animals were intravenously injected with GSNO (10 mM in saline, 0.1 mL kg$^{-1}$) and GSH (30 mM in saline, 0.1 mL kg$^{-1}$), and intramuscularly with benzylpenicillin sodium salt solution (20,000 u mL$^{-1}$, 0.5 mL kg$^{-1}$) every day for 7 consecutive days since stent implantation. At 1 week, 1 month, and 3 months post implantation, the stented arteries ($n = 12$ for each group at each time point) were explanted and the animals were euthanized with excess pentobarbital sodium salt solution.

**Vascular stent deployment in pigs**. Four types of vascular stents were compared in the preclinical trial in pigs. All of them were prepared with the same type of CoCr BMS manufactured by Kossel Medtech as aforementioned to avoid the influence of different stent designs on the outcome. NOE hydrogel and blank hydrogel-coated stents were prepared by ourselves as aforementioned, while DES and polymer-coated stent were produced by this company. Poly(vinylidene fluoride-co-hexafluoropropylene) (PVDF-HFP) was utilized as the polymer coating and everolimus was assumed as the anti-restenotic drug (8.5 ± 2.1 µg mm$^{-1}$).

A total of 12 ($n = 6$ for each time point) male *Bama* miniature pigs (~6 months old, ~30 kg) were used for this study. The pigs had been fed with clopidogrel (75 mg) and aspirin (100 mg) every day since 3 days prior to stent deployment and were fasted overnight before it. Each animal was sedated with Zoletil® 50 (50 mg mL$^{-1}$, 5 mg kg$^{-1}$; Virbac, France) and ketamine (50 mg mL$^{-1}$, 10 mg kg$^{-1}$). Afterward, it was intubated and then connected to a ventilator (DM-6A, Superstar Medical Equipment, China). Anesthesia was maintained with isoflurane (2 v/v%) throughout the procedure. After administering heparin (200 U kg$^{-1}$) and antibiotic prophylaxis intravenously, the right femoral artery was exposed under sterile condition. Four vascular stents (one for each type) were randomly implanted into three to four coronary arteries through a 6-Fr catheter under the guidance of DSA (Innova IGS 530, GE Healthcare, USA). The stents were dilated till the stent-to-artery diameter ratio reaching ~1.1 and the pressure was maintained for 30 s. The animals were also intravenously injected with GSNO (10 mM in saline, 0.1 mL kg$^{-1}$) and GSH (30 mM in saline, 0.1 mL kg$^{-1}$) every day for 7 consecutive days since stent implantation. At 2 weeks and 3 months post implantation, the stented arteries were explanted after the patency of them had been checked with DSA. In the end, the animals were euthanized with excess potassium chloride solution at anesthesia.

**Characterization of the stented arteries**. The harvested stented arteries were extensively washed with heparin solution (50 U mL$^{-1}$) and then cut transversely into two segments. One-half of the segments was fixed in paraformaldehyde solution while the other half of them was further cut lengthwise and then fixed in paraformaldehyde or glutaraldehyde solution.

The columnar segments in paraformaldehyde solution were further cut transversely, and then dehydrated by gradient alcohols and desiccation in air. Most of them were cured in methyl methacrylate (with 2 w/v% benzoyl peroxide, 50 °C, 24 h). Afterward, the solidified resins were sliced by a hard-tissue microtome (BQ1600, Lan Ming Medical Treatment, China). The generated tissue sections were treated by a van Gieson staining kit (Cat. No.: abs9349, Absin Bioscience, China) following the protocol provided by the vendor, and then photographed using an optical microscope (CKX41, Olympus, Japan; software: ToupView 2.4). The generated images were analyzed with ImageJ 1.52a (NIH, USA). The metal stents of the remaining segments were electrolyzed according to the protocol provided by Veinot et al.[53] Afterward, paraffin sections of them were prepared and then treated by either hematoxylin and eosin (H&E) staining or CD68 immunostaining.

The semi-columnar segments in paraformaldehyde solution were further permeabilized with Triton X-100 solution for 3 h and then blocked with BSA solution for 12 h. Subsequently, they were incubated with mouse anti-rabbit CD31 antibody (IgG, Cat. No.: NBP2-44342, Novus Biologicals, USA, 1/100 dilution in PBS) or mouse anti-porcine CD31 antibody (IgG, Cat. No.: NB100-65336, Novus Biologicals, USA, 1/100 dilution in PBS) for 6 h, and then Alexa Fluor® 488-conjugated goat anti-mouse IgG secondary antibody (Cat. No.: abs20013, Absin Bioscience, 1/100 dilution in PBS) for another 6 h. After that, they were simultaneously stained with phalloidin-TRITC and DAPI for 6 h. Finally, the samples were photographed using a CLSM (A1 Plus, Nikon, Japan; software: Nikon NIS-Elements AR 5.01.00) after being extensively washed with PBS. The generated images were analyzed with Bitplane Imaris 9.0.1 (Oxford Instruments, UK). The remaining semi-columnar segments in glutaraldehyde solution

were dehydrated by gradient alcohols, dried in air, sputter-coated with gold, and then examined by SEM.

**Statistical analysis**. One-way analysis of variance (ANOVA) with Tukey post hoc test was performed for comparison among multiple groups. Independent two-tailed student's *t*-test was assumed to determine the difference between two groups in some cases.

**Ethical issues on animal experiments**. All procedures on the animals were approved by the Institutional Animal Care and Use Committee of Sun Yat-Sen University and by the Committee on Ethics in Medical Research of Southwest Jiaotong University.

**Reporting summary**. Further information on research design is available in the Nature Research Reporting Summary linked to this article.

## Data availability

The authors declare that the data supporting the findings of this study are available within the paper and its supplementary information files. The source data generated in this study and raw data for the transcriptome analysis are available in Zenodo database at https://zenodo.org/record/5611259#.YXtZkp5Bw2x. Additional data are available from the corresponding authors upon reasonable request. The list of figures that have associated source data: Fig. 1b, Fig. 1c, Fig. 1d, Fig. 1e; Fig. 2a, Fig. 2b; Fig. 3b, Fig. 3c, Fig. 3d, Fig. 3e, Fig. 3f; Fig. 4b, Fig. 4d, Fig. 4e, Fig. 4f, Fig. 4h, Fig. 4i; Fig. 5a, Fig. 5b, Fig. 5c, Fig. 5d, Fig. 5e, Fig. 5g; Fig. 6c, Fig. 6d; Fig. 7e, Fig. 7f, Fig. 7g.

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

## Acknowledgements

This work was financially supported by the National Natural Science Foundation of China (No. 32000939 to Y. C., No. 22004135 to L. H., and No. 82072072 to Z. Y.), National Key R&D Program of China (No. 2017YFE0102400 to J. Z.), General Research Fund from the Research Grants Council of Hong Kong (Nos. 16308818 and 16309920 to H. W.), Shenzhen Fundamental Research Program (No. JCYJ20190807160415074 to Y. C., and No. JCYJ20190807160401657 to J. Z.), International Cooperation Project by Science and Technology Department of Sichuan Province (No. 2021YFH0056 to Z. Y.), and Shenzhen Science and Technology Program (No. 2021A15 to L. H.). We gratefully acknowledge Ms. T. You from MOE Key Laboratory of Advanced Technologies of Materials at SWJTU for the help with SEM, and Ms. Z. Hu from the Analytical and Testing Center at SWJTU for the aid with CLSM. We appreciate Mr. E. M. W. Fok from the Department of Chemistry at HKUST for ICP-MS analysis. We also thank Mr. W. Huang and Ms. Z. Xiao from the School of Biomedical Engineering at SYSU for the artwork.

## Author contributions

Y.C., Z.Y. and H.W. conceived the idea for this study. Y.C. designed, prepared and characterized the materials with the aid of X.D., Y.C. and L.H. undertook the in vitro study. Y.C., P.G., X.T., N.Z., T.Y. and H.Q. carried out the ex vivo and in vivo experiments. Y.C. performed the transcriptome analysis with the aid of P.G., S.M., Q.T., N.H. and Z.G. contributed technical supports. Y.C., P.G., L.H., J.Z., Z.Y. and H.W. analyzed the data and discussed the results. Y.C. and J. Z. wrote and edited the paper. J.Z., Z.Y. and H.W. supervised the study. All authors participated in revising the paper and agreed on the final version of it.

## Competing interests

The authors declare no competing interests.
