## [Peer Review File · Nature Communications]

The previous round of reviews was done at another journal

Response to reviewers' comments for "NBME-19-1512B-Z"

Reviewer #1 (Report for the authors (Required)):

Excellent translational work! Great job in addressing the comments. Also, the additional swine model study including DES and the presentation of new in vitro data increase the impact of the work.

A brief summary: A tough 'endothelium-like dressing' tough hydrogel is developed here to target the in-stent restenosis (ISR) issue of stents. Convincing in vitro and in vivo studies demonstrated that the dressing promoted rapid formation of a native endothelium on vascular stents and suppressed ISR. Overall, this work represents the state of the art, having translational significance. The studies on the rabbit and swine models provide nice preclinical data for future translation advancements.

Everything is clearly and nicely presented. The overall manuscript could be made more concise.

Reply: Dear reviewer, thanks a lot for your positive comments! We have made our manuscript more concise in our newest version.

Reviewer #2 (Report for the authors (Required)):

The authors have responded to many of the inquiries and performed new experiments to highlight the functionality of their NO releasing modified alginate coating. The material science is well documented and there are nice aspects of the work that document a functional activity of the coating. Yet, the idea that these data now tell a compelling story for a novel drug eluting stent and indeed one that is endothelial like is still problematic.

1 ENDOTHELIAL-LIKE

I would still respectfully but firmly note that the device presented is not an endothelium or endothelial-like and any reference to endothelial-like should be removed from the title and paper. There is no proof provided that this the coating has any endothelial like functionality. This is a coated stent that elutes NO and functions as well like other drug-eluting stents. Calling it endothelium or endothelial-like implies that it is responsive in a manner emblematic of a living dynamic system and functions in the way a cellular coating should function. Moreover, it implies that the coating is intact and endothelial-like even before implantation and not that the coating has favorable reparative properties that fosters endothelial recovery. This is simply not the case here. A number of coatings elute a drug, are thromboresistant and can enable endothelial recovery and this class of coatings is never referred to as endothelial-like. Any mention of being endothelial-like with this device is frankly misleading and conveys a very different message than what is presented here.

Reply: Dear reviewer, thanks a lot for your comments! We agree that our coating is not a living coating and cannot function exactly like a native endothelium. We respect your opinion and renamed it as “endothelium-inspired (EI) dressing”. Nevertheless, we believe that our coating does provide some endothelial functions, as NO plays critical roles in nearly all important biological functions of native endothelium. In fact, other important signaling molecules such as prostacyclin, thrombomodulin and heparin-like molecules could be loaded in our coating. However, incorporating all important biomolecules generated by native endothelium into it will make our study too complicated to be implemented.

2 ALGINATE MALEIMIDE (A-M) AND GELATIN MATERIAL

The material presented seems more durable than similar alginate coatings used in the past. This new material is still degradable though I imagine. If so over how long a period of time? Presumably more than one month. FDA guidelines and accepted procedures require a multiple of the degradation time before declaring safety. One month data are interesting and follow expected course. What does the response look like at longer time points, e.g. three times the degradation time? This is especially important as the NO release is complete after 2+ weeks

Reply: Our EI dressing is biodegradable and it might have been replaced by the natural extracellular matrix secreted by the adhering cells in three months. Nevertheless, the drug-releasing data (Supplementary Fig. 44) suggest the EI dressing could last for at least one month. According to our observation, an intact native endothelium had regenerated on EI dressing-coated stents in two weeks, which would eventually replace the temporary EI dressing.

3. STENT COMPARISON

I am a bit confused – it appears that the porcine experiments all used the same cobalt chrome stent backbone - was this the case for the rabbit? What were the devices used in the rabbit? Are they the same as the porcine? Please provide more information on stent strut dimensions. Dimensions of the struts are essential in comparing devices and the devices used here seem to have a high strut to artery surface area ratio.

Reply: For the rabbit experiment, the stent backbone is made of 316L stainless steel. The width and thickness of the struts are 97 μm and 100 μm , respectively. The strut-to-artery surface area ratio of it is 0.153. For the swine experiment, the stent backbone is made of cobalt chromium alloy (CoCr). The width and thickness of the struts are 85 μm and 89 μm , respectively. The strut-to-artery surface area ratio of it is 0.147. We have included this information in our modified manuscript.

4-5. THROMBOSIS MODELS

Again here I want to be clear that I understand – what is reported is this is the thrombogenicity of flat stainless-steel foils that are uncoated or coated and speaks to the material coating as anti-thrombotic but not devices employed in the animal experiments.

4 If this is the case then this needs to be made clear

Reply: To demonstrate that the EI dressing-coated stent is antithrombotic, we compared the bare-metal stent (BMS) and hydrogel-coated stents using the *ex vivo* thrombogenicity test. Our result shows that both BMS and blank hydrogel-coated stents are highly thrombogenic, while the EI dressing-coated stents could effectively retard blood coagulation or nearly non-thrombogenic depending on the content of conjugated SeCA (see Supplementary Fig. 28). Such result is similar to that of foils. We used foils to evaluate the thrombogenicities of our hydrogel coatings due to the high cost of the vascular stent.

5 please provide the thrombotic events in animal models – it seems like they are comparable.

Reply: Sorry we didn't examine the thrombotic events because all four groups were tested on the same rabbits to minimize the systemic errors. Even if there were thrombotic events, we were unable to tell which groups of substrates caused them.

6. Endothelial recovery data

Please present the data on endothelial recovery itself – it is hard to see what was called endothelial recovery. The images provided are nice and show what appears to be a cellular monolayer which is morphologically different for different devices but this does not correlate with extent of thrombosis or degree of intimal hyperplasia. Was there such direct association?

Reply: Thanks a lot for your kind suggestion! We have quantified the degree of endothelial coverage (Fig. 7f) and correlated the mean endothelial coverage between 2 weeks and 3 months post stent deployment with the increment in neointimal thickness during that period (Fig. 7g).

Reviewer #3 (Report for the authors (Required)):

The present manuscript describing an endothelial like coating on vascular stents composed of hydrogel composed of alginate and gelatin, which are analogs to hyaluronic acid and collagen in extracellular matrix. The authors report inhibition of smooth muscle cell growth and proliferation with minimal impairment of endothelial regrowth using an animal model of stenting in the rabbit and pig coronary. While the authors have done a good job of responding to my criticisms, several issues remain with the manuscript in its current state.

1) The assays in Figure 4 have been improved somewhat but I am concerned about the specific assays done and their interpretation. Line 327. page 15 (Quantitative analyses revealed no significant difference in the proliferation coverage and spreading on HUVECs among the ... "In the Figure 4c (which I assume is the basis for data in d-f) it appears the SS had earlier and better coverage by HUVECs but yet only cell coverage shows a clear difference at 3 days? Cell density looks different. Another problem with this assay is that of course everything will look the same at 7 days--there is only so much room for cells to grow because they become contact inhibited. Also there is a comment about no difference in cell proliferation rate when only viability was measured-these is not the same as proliferation."

Reply: Dear reviewer, thanks a lot for your comments! There does be no significance of difference in cell density among all groups at a given time. To present this more clearly, we displayed only the cell nuclei in Figure 4c as below. Figure 4h (relative viability) has nothing to do with endothelial cells. It shows the viability of smooth muscle cells after co-cultured with the blank hydrogel or EI dressings containing varying contents of SeCA in the presence of GSNO.

Figure 1. Confocal laser scanning microscopy images displaying the proliferation of HUVECs seeded onto various substrates in the presence of GSNO. Only cells nuclei (blue) were stained (Scale bar: 100 μ m)

2. I am lost on line 375 page 17 "However, the EL dressings were unable to stop the proliferation of smooth muscle cells due to the unsustainable generation of NO *in vitro*. Nonetheless, this may not be an issue *in vivo* since the volumes of blood in experimental rabbits and pigs are two to three magnitudes larger." What is this referring to?

Reply: We feel very sorry for the confusion caused by us! For *in vitro* study, the volume of medium was only 1 mL so that GSNO (the NO donor) dissolved in such a small amount of liquid would have been used up in one hour according to our estimation. However, for *in vivo* study, the blood volumes were 125~200 mL for the rabbits, and 1.25~2 L for the pigs, respectively. Consequently, GSNO dissolved in the whole blood of the animals was enough to sustain NO generation for many days. We have made it clearer in our revised manuscript.

3. The transcriptome explanation is uninterpretable. No clear message but an indicated things went more up than down. Please pick some specific messages this data is trying to convey and be clear which transcripts support it and which do not. There is a lot of terms used like regulated inflammation which really aren't very specific. Similar for innate inflammatory and pro-inflammatory. This whole experiments adds little to me and is not very convincing.

Reply: We are sorry for the confusion caused by us! We have simplified our discussion on this part and focused on the most relevant message conveyed by the transcriptome analysis. Thanks a lot for the kind suggestion!

4. Vascular stents in rabbit. I don't see any SD on the data in reference to neointimal growth which is where the authors claim the benefit is? Are those differences in neointimal growth significant between timepoints? n=? Where is this data shown graphically? The term ISR stands for in stent restenosis and is used to refer to stents with 50% diameter stenosis. Perhaps the authors mean % cross sectional area? This part is really unclear. The data are presented poorly and I am not convinced they mean anything. Figure 6d-can we get a percentage of coverage of CD31 instead of fluorescence readout? Over strut would be more important here. I don't see quantitation of SEM data?

Reply: We are confused by question 1 and question 4. Figure 6b has already shown the cross-sections of representative stented arteries at 1 week, 1 month, and 3 months post stent implantation for both the bare-metal stent and EI dressing-coated stent. The error bars in Figure 6c represent SD in reference to neointimal growth. n = 12 for each group at each time point has been indicated in the figure legend as well. There is significant difference in neointimal growth among timepoints for both types of stents. We showed that in Supplementary Fig. 30 in the revised Supplementary Information. We have changed the term 'in-stent restenosis' to 'area stenosis' to avoid misleading. SEM is not accurate to reflect the degree of endothelial coverage. However, we have taken CLSM images on multiple specimens for each type of stent and quantified its degree of endothelial coverage (see Fig. 7f and Supplementary Fig. 31).

5. Pig experiments. Page 26 last line--it says severe narrowing occurred in 3/6 polymer coated stent yet the histology does show this--why? Is supplementary video 4 polymer only? It appears with sever ISR but the histomicrograph does not correlate? Why? What polymer was used in the polymer only stent? I think see a thick neointima in the polymer stent but it appears to be greater than in the EL dressing and DES. What DES

was used? The strut diameter used in all animal experiments is not consistent with what is clinically used--why? Was the DES commercially available? Again the use of ISR is incorrect. Do you mean area stenosis? What is most impressive is the difference between the blank hydrogel and the polymer only--why polymer only as a control? You used BMS in the last experiment? IN 7F, do we have quantitation for this over the whole stent.

Reply: The segments of stented arteries for histological study were not properly handled by ourselves in the beginning. Fortunately, we had preserved a segment for each specimen. We sent them to professionals for dehydration, embedding, sectioning and staining. We have replaced the old pictures with new ones, which correlate well with DSA images now. In Supplementary Video 4, a polymer-coated stent was implanted in the left circumflex (LCX) artery while an EI dressing-coated stent was implanted in the left anterior descending (LAD) artery. For the polymer-coated stent, poly(vinylidene fluoride-co-hexafluoropropylene) (PVDF-HFP) was utilized as the coating. For the DES, everolimus was assumed as the anti-restenotic drug and loaded in the polymer coating ($8.5 \pm 2.1 \mu\text{g mm}^{-1}$). The width and thickness of the struts are 85 μm and 89 μm , respectively. The strut-to-artery surface area ratio of it is 0.147. Our DES is comparable to Xience V[®], Xience Prime[®], Promus Element[®], and Promus Premier[®]. All the stents (including our DES) used for porcine study were prepared with the same type of CoCr BMSs manufactured by Kossel Medtech to avoid the influence of different stent designs on the outcome. We have included the relevant information in our revised manuscript. We have changed the term 'in-stent restenosis' to 'area stenosis' to avoid misleading. In the rabbit experiment, we have already shown the advantage of our EI dressing-coated stent over BMS. We compared the blank hydrogel-coated stent with polymer-coated stent head to head in order to demonstrate that even the blank hydrogel coating is more biocompatible than the non-biodegradable polymer coating. We have taken CLSM images on multiple specimens for each type of stent and quantified its degree of endothelial coverage (see Fig. 7f and Supplementary Fig. 31).

6. The *in vivo* data are critical to prove the authors' conventions about the EL coating but yet the data on endothelialization is not completely presented using confocal and SEM?

Reply: SEM is not accurate to reflect the degree of endothelial coverage. We wished we could image the entire stent with CLSM, but we were unable to achieve that because of two technical difficulties. The first one is that the sample preparation procedures included sandwiching the stent between a glass slide and a glass coverslip, during which the endothelium at some regions was stripped off. The second one is that each image was reconstructed as the maximal projection of tens of Z-stack images, which took nearly one hour due to the low scanning speed of the instrument. It was impractical for us to scan the entire stent because it would take a few days for each specimen while the instrument is for public use in the whole university. However, we have taken CLSM images on multiple specimens for each type of stent and quantified its degree of endothelial coverage (see Fig. 7f and Supplementary Fig. 31).

7. There are statements about inflammation and polymer page 28 which are speculation and not back by data. I don't see any data on inflammation. You didn't look at endothelial function in the *in vivo* experiments so how can you comment on it?

Reply: We electrolyzed the metal struts in the columnar segments of stented arteries and prepared paraffin sections of them for Hematoxylin and Eosin (H&E) Staining or CD68 immunostaining. For rabbit study, our results (Supplementary Fig. 34 to 37) demonstrated BMS induced moderate to severe inflammation (see Supplementary Table 1 for the classification of inflammation) within 1 month, though the degree of inflammation reduced slightly after 3 months. In stark contrast, our EI dressing-coated stent only induced

minimally inflammation during implantation. For porcine study, our results (Supplementary Fig. 41 to 43a) revealed polymer-coated stent induced severe inflammation during deployment while mild to moderate inflammation was observed on other stents. Notably, DES effectively suppressed inflammation within 2 weeks. However, such repressive effect was unsustainable as the inflammation in the stented arteries exacerbated after 3 months. In contrast, EI dressing-coated stent persistently suppressed inflammation during 3 months. Although blank hydrogel-coated stent induced moderate inflammation in the beginning, such inflammation mitigated after 3 months.

For porcine study, we plotted the increments in neointimal thickness between 2 weeks and 3 months post stent deployment versus the mean endothelial coverage during that period of time (Fig. 7h). Our data suggested NIH was negatively correlated the degree of endothelialization with this duration. In particular, neointimal growth was strongly suppressed on EI dressing-coated stent for which an intact endothelium had formed at 2 weeks post deployment.

REVIEWER COMMENTS

Reviewer #2 (Remarks to the Author):

The authors have done a decent job of responding to my comments. I do think it would be appropriate for the authors to point out some of the limitations of their current work. Their stent was tested against non other stents not approved for human use. Thus many of the comparators are not really ideal comparisons given we are not given any information about quality control etc. Notably PVDF is usually well contracted coating and the results in the pig are not representative of my experience with it. please tone your claims down and acknowledge the limitations of your study

Reviewer #3 (Remarks to the Author):

The authors have revised their paper and have responded to many of the comments – yet at the same time I find the primary message of this work clouded by seemingly extraneous material and insistence on using terms and words that have different meanings to communities (e.g. vascular biologists etc.).

Title: there are multiple issues with the title

1 the title begins with TOUGH – this implies that the material is excessively durable and in fact this has not been shown – the composite alginate-gelatin is tougher than alginate alone. But a tensile strength of kPa is nowhere near what we would consider as tough material. This is not a tough material and references in the title to toughness is misleading. kPa tensile strength cannot compare to the MPa toughness of other hydrophobic stent coatings.

2 the authors have changed from endothelial like to endothelial inspired but really should remove this entirely. It is misleading to use the term Endothelial Inspired. This is an NO generating hydrogel material and has no aspect of the endothelium. I would suggest removing all but for example one reference in the discussion that amongst the many things the endothelium does is regulate NO biology. Such a title will leave the implication to the reader that this material has endothelial like properties and it does not. Moreover, if this is true here it would be true for any of the multitude of releasing or generation of compounds that are associated with the endothelium. Please remove endothelial inspired from the title and text.

3 the use of the term dressing is similarly misleading – and on line 951 use of the term “vascular stent encapsulated by the EI dressing” implies that the dressing is apart from the stent and wrapped around it rather than strut adherent. On lines 912-918 it seems like the stents were coated with a base layer and then dip coated. This would imply that there is a coating that is an integral part of the stent and not a separate material. Dressing implies that material is separate.

The title should be revised to declare what has been reported – i.e. An alginate-gelatin nitric oxide generating coating reduces stent intimal hyperplasia

4 Line 54 Endothelium-mimetic

The coating here is not endothelium-mimetic, it generates NO. There are multiple properties of the endothelial and multiple endothelial products. To imply that hydrogels are endothelial-mimetic or that inclusion of an NO generating compound is endothelium-mimetic is simply wrong.

5 Line 66 alginate-gelatin

Alginate gelatin hydrogels are widely used see e.g. Sun et al BBRC 2016, 477 1085 and others including ref. 18. This needs to be made clear and for that matter given what has been published in this area

6 lines 107-243 and fig 2 are as noted above extraneous and can be removed or reduced to a few lines

All of this discussion and figure 2 seems extraneous and not relevant to the paper as there is only minimal difference in materials strength and the mechanical properties of these composite materials is well addressed in the literature. One can simply include in the text reference to the materials properties of what was used and remove all discussion of materials not employed in the in vivo studies. These other materials are a distraction and add nothing to the paper as the materials properties seemingly had no impact on the biologic response/

7 lines 313-316 and Figure 4

The idea that the blank hydrogel coating selectively facilitated the adhesion of endothelial cells, while NO generation catalyzed by the dressings further inhibited the attachment of smooth muscle cells does not seem to be supported by the data. Figure 4 d-f seems to show no difference between the alginate alone and with NO generation. Moreover, 4.b shows no effect of NO generation on endothelial density and no dose effect and none of the formulations reaching complete coverage. This figure calls into question many of the claims of endothelial recovery and reduction of smooth muscle cell adhesion.

8 lines 347-349

I do not see the data to support the idea that a confluent monolayer is formed. Islands of connected endothelial cells is not a single confluent monolayer

9 lines 348-360

If I understand correctly cell viability was the number of living cells relative to total cells in the area and to claim an effect given the incredibly modest difference does not do justice to the biology of the cells. A statistical difference over a range equivalent to the variance for an individual group is not likely to have a biological effect.

10 Fig 5

The PCA analysis shows minimal variability and I do not see how this can be biologically relevant

11 lines 460-462

316L stainless steel stents are NOT widely used in clinical practice

12 endothelial coverage

I am not clear as to how endothelial coverage was assayed for in the animals. Seems like IHC was used with antiCD31 antibodies. Was this characterized with confocal microscopy or en face imaging?

13 thrombosis

I do not understand the idea of not reporting thrombosis on stents implanted in left and right iliac arteries or in the different coronary arteries of the pig. Are you saying that the placement of an uncovered stent in one iliac or coronary will have impact on the other. If this is the case for thrombosis, it should be the case for inflammation, endothelial recovery and intimal hyperplasia as well. Were there any thrombotic events in any of the animals

14 animal studies

Did the results vary on which coronary artery was used

15 animal studies

Seems like what can be said is that NO generated coatings reduce intimal hyperplasia and have early endothelial coverage comparable to standard drug eluting stents. The NO generating coatings have greater endothelial coverage but more intimal hyperplasia early than clinical drug-

eluting stents which at three months show no difference in endothelial coverage and slightly more intimal hyperplasia.

In short, this paper is way too long including information that distracts from the primary message – i.e. that an NO generating coating is effective in two animal models at reducing intimal hyperplasia akin to release of an intact compound. The central idea is buried in a lot of extraneous material. The paper is then simultaneously is far too long and contains too much information of no value and then does not delve into the link between potentially innovative chemistry and vascular biology. What is presented should be reduced significantly and what is not should be expanded. Entire sections could be removed AND make the paper more readable.

REVIEWER COMMENTS

Reviewer #2 (Remarks to the Author):

The authors have done a decent job of responding to my comments. I do think it would be appropriate for the authors to point out some of the limitations of their current work. Their stent was tested against non other stents not approved for human use. Thus many of the comparators are not really ideal comparisons given we are not given any information about quality control etc. Notably PVDF is usually well contracted coating and the results in the pig are not representative of my experience with it. please tone your claims down and acknowledge the limitations of your study.

Reply: Thanks for your suggestion! We have discussed the limitations of our study at the "Discussion" section in the revised Manuscript, and toned down the claims made by us.

Reviewer #3 (Remarks to the Author):

The authors have revised their paper and have responded to many of the comments – yet at the same time I find the primary message of this work clouded by seemingly extraneous material and insistence on using terms and words that have different meanings to communities (e.g. vascular biologists etc.).

Title: there are multiple issues with the title

1. The title begins with TOUGH – this implies that the material is excessively durable and in fact this has not been shown – the composite alginate-gelatin is tougher than alginate alone. But a tensile strength of kPa is nowhere near what we would consider as tough material. This is not a tough material and references in the title to toughness is misleading. kPa tensile strength cannot compare to the MPa toughness of other hydrophobic stent coatings.

Reply: The word “tough” in our article title implies that our hydrogel is much tougher than conventional hydrogels. We were not comparing our hydrogel with plastic materials or other materials. In our case, we have compared it with pristine gelatin hydrogel, UV-crosslinked gelatin methacrylate (GelMA) hydrogel, and alginate dialdehyde/gelatin (A-D/G) hybrid hydrogel. The results (Supplementary Fig. 12, Supplementary Fig. 21 and Supplementary Fig. 23 in the previous Supplementary Information) showed the mechanical property of our hydrogel was much better than those of others. In the highly cited article “Highly stretchable and tough hydrogels” (*Nature* **2014**, 489, 133), the authors claimed their hydrogels to be highly stretchable and tough. The maximal tensile strength of their hydrogels was only twofold as that of ours. Therefore, we believe it is reasonable to say that our hydrogel is tough. However, to avoid misleading, we revised our article title to be “A tough nitric oxide-eluting hydrogel coating suppresses neointimal hyperplasia on vascular stent”.

2. The authors have changed from endothelial like to endothelial inspired but really should remove this entirely. It is misleading to use the term Endothelial Inspired. This is an NO generating hydrogel material and has no aspect of the endothelium. I would suggest removing all but for example one reference in the discussion that amongst the many things the endothelium does is regulate NO biology. Such a title will leave the implication to the reader that this material has endothelial like properties and it does not. Moreover, if this is true here it would be true for any of the multitude of releasing or generation of compounds that are associated with the endothelium.

Please remove endothelial inspired from the title and text.

Reply: Thanks for your comments! We have removed the words “endothelium-inspired” entirely, and revised our article title to be “A tough nitric oxide-eluting hydrogel coating suppresses neointimal hyperplasia on vascular stent”.

3. The use of the term dressing is similarly misleading – and on line 951 use of the term “vascular stent encapsulated by the EI dressing” implies that the dressing is apart from the stent and wrapped around it rather than strut adherent. On lines 912-918 it seems like the stents were coated with a base layer and then dip coated. This would imply that there is a coating that is an integral part of the stent and not a separate material. Dressing implies that material is separate. The title should

be revised to declare what has been reported – i.e. An alginate-gelatin nitric oxide generating coating reduces stent intimal hyperplasia.

Reply: Thanks for your comment and suggestion! We have used “nitric oxide-eluting (NOE) hydrogel (coating)” to replace “endothelium-inspired (EI) dressing”, and used “coated with” to replace “encapsulated by” in the revised Manuscript and Supplementary Information. The article title has also been changed to “A tough nitric oxide-eluting hydrogel coating suppresses neointimal hyperplasia on vascular stent”.

4. Line 54 Endothelium-mimetic

The coating here is not endothelium-mimetic, it generates NO. There are multiple properties of the endothelial and multiple endothelial products. To imply that hydrogels are endothelial-mimetic or that inclusion of an NO generating compound is endothelium-mimetic is simply wrong.

Reply: Thanks for your comments! We have used “nitric oxide-eluting (NOE) hydrogel (coating)” to replace “endothelial-inspired (EI) dressing” in the revised Manuscript and Supplementary Information.

5. Line 66 alginate-gelatin

Alginate gelatin hydrogels are widely used see e.g. Sun et al. *BBRC* 2016, 477 1085 and others including ref. 18. This needs to be made clear and for that matter given what has been published in this area.

Reply: Our alginate maleimide/gelatin (A-M/G) hybrid hydrogel is a new material, which has not been reported before. Sun et al. developed a silk fibroin/collagen hybrid material for composite scaffolds (*Biochem. Biophys. Res. Commun.* **2016**, 477, 1085). We cannot find the word “alginate” or “gelatin” in their article. Reference 18 (*Nat. Rev. Mater.* **2018**, 3, 159) is a review paper, we cannot find the word “gelatin” in it as well.

6. Lines 107-243 and fig 2 are as noted above extraneous and can be removed or reduced to a few lines

All of this discussion and figure 2 seems extraneous and not relevant to the paper as there is only minimal difference in materials strength and the mechanical properties of these composite materials is well addressed in the literature. One can simply include in the text reference to the materials properties of what was used and remove all discussion of materials not employed in the in vivo studies. These other materials are a distraction and add nothing to the paper as the materials properties seemingly had no impact on the biologic response.

Reply: Thanks for the comment and suggestion! In that two sections, we demonstrated that our hydrogel coating could maintain integrity on the vascular stent while weak hydrogel coatings fractured or even peeled off it during balloon dilation. The good mechanical property of our hydrogel is very important though it may have no effect on biology. If it were a weak hydrogel coating on a vascular stent, it would fracture during the angioplasty process and its debris might induce embolization in downstream capillaries, causing micro-infarctions.

Our hydrogel is a new material so that we had to use great lengths of text to describe the mechanical property of it. The elucidation on the mechanism of toughness is a highlight of this study. It is helpful for interpreting the mechanical behavior of our hydrogel and the design of new

hydrogels with better mechanical performance. We think preserving it will attract more attention from the readers of material science.

Nevertheless, we agree that there are some extraneous results and discussion. To avoid distraction from the main topic, we have removed them from the revised Manuscript and Supplementary Information. For instance, the comparison between alginate-maleimide/gelatin (A-M/G) hydrogel and alginate-dialdehyde/gelatin (A-D/G) hydrogel was thoroughly removed.

7. Lines 313-316 and Figure 4

The idea that the blank hydrogel coating selectively facilitated the adhesion of endothelial cells, while NO generation catalyzed by the dressings further inhibited the attachment of smooth muscle cells does not seem to be supported by the data. Figure 4 d-f seems to show no difference between the alginate alone and with NO generation. Moreover, 4.b shows no effect of NO generation on endothelial density and no dose effect and none of the formulations reaching complete coverage. This figure calls into question many of the claims of endothelial recovery and reduction of smooth muscle cell adhesion.

Reply: We compared the density of smooth muscle cells on the blank hydrogel and the nitric oxide-eluting (NOE) hydrogels in a separate figure as below (Fig. R1). Our data do show that NO generation catalyzed by the NOE hydrogel further inhibited the attachment of smooth muscle cells.

Fig. R1 | Quantitative analysis on the adhesion of human umbilical artery smooth muscle cells (HUASMCs) on the blank hydrogel and nitric oxide-eluting (NOE) hydrogels containing varying contents of selenocystamine (SeCA). The cell growth medium was supplemented with S-nitrosoglutathione (GSNO, 10 μ M) and glutathione (GSH, 30 μ M). One-way analysis of variance (ANOVA) with Tukey post-hoc test was performed to determine the difference among various substrates. (** $P < 0.01$, *** $P < 0.001$ and **** $P < 0.0001$).

Our results (Fig. R1 and Fig. 4a-4b (i.e. Fig. R2)) also support the statement “the blank hydrogel coating selectively facilitated the adhesion of endothelial cells, while NO generation catalyzed by the dressings further inhibited the attachment of smooth muscle”. There is no significant difference in the adhesion of human umbilical artery smooth muscle cells (HUASMCs) between stainless steel and the blank hydrogel. In contrast, the number of human umbilical artery smooth muscle cells (HUASMCs) on the blank hydrogel was less than half of that on stainless steel.

Fig. R2 (i.e. Fig. 4a-4b) | Effects of the nitric oxide-eluting (NOE) hydrogels on cellular behaviors *in vitro*. **a**, Fluorescence images exhibiting the competitive adhesion between human umbilical vein endothelial cells (HUVECs) and human umbilical artery smooth muscle cells (HUASMCs) on various substrates. The cell growth medium was supplemented with S-nitrosoglutathione (GSNO, 10 μ M) and glutathione (GSH, 30 μ M). (scale bar: 500 μ m) **b**, Quantitative analyses on the competitive adhesion between HUVECs and HUASMCs ($n = 6$). One-way analysis of variance (ANOVA) with Tukey post-hoc test was performed to determine the difference among various substrates and two-tailed Student's *t*-test was assumed to determine the difference between the two types of cells on the same substrate. (##### $P < 0.0001$ compared to other groups; ** $P < 0.01$ and **** $P < 0.0001$).

Fig. 4d-4f (i.e. Fig. R3) in the Manuscript were related to the adhesion, proliferation and spreading of HUVECs. The purpose of these three figures is to demonstrate HUVECs could adhere, spread and proliferate on our hydrogels, and NO generated from the nitric oxide-eluting hydrogels had no detrimental effects on HUVECs. To avoid misleading, we have inserted "HUVECs" in the figure legend. We did not claim that NO had any effect on the behavior of endothelial cells.

Fig. R3 (i.e. Fig. 4d-4f) | Summary of cell density, cell coverage and individual cell area for human umbilical vein endothelial cells (HUVECs) on various substrates ($n = 6$).

The competitive cell adhesion test of HUVECs and HUASMCs on various substrates was performed at a total seeding density of only 10,000 cells cm⁻² for 3 h. The small number of endothelial cells were unable to reach complete coverage in 3 h.

8. Lines 347-349

I do not see the data to support the idea that a confluent monolayer is formed. Islands of connected endothelial cells is not a single confluent monolayer.

Reply: Herein we wanted to express that the cells formed confluent monolayers in 1 week (as shown in Fig. 4c (i.e. Fig. R4)). We have included the time information in the revised Manuscript.

Fig. R4 (i.e. Fig. 4c) | Confocal laser scanning microscopy (CLSM) images displaying the adhesion, spreading and proliferation of human umbilical vein endothelial cells (HUVECs) seeded onto various substrates in the presence of S-nitrosoglutathione (GSNO). (scale bar: 100 μ m)

9. Lines 358-360

If I understand correctly cell viability was the number of living cells relative to total cells in the area and to claim an effect given the incredibly modest difference does not do justice to the biology of the cells. A statistical difference over a range equivalent to the variance for an individual group is not likely to have a biological effect.

Reply: Thanks for the comment! We used CCK-8 to measure the proliferation of the smooth muscle cells (Fig. 4h (i.e. Fig. R5)). We agree that “viability” is not accurate to describe their proliferative capability. Therefore, we have replaced “viability” with “vitality” in the revised Manuscript now. With respect to statistical difference, the data were analyzed by one-way analysis of variance (ANOVA) with Tukey post-hoc test with $n = 8$. The conclusion on the significance of difference should make sense.

Fig. R5 (i.e. Fig. 4h) | h, Proliferation assay of human umbilical artery smooth muscle cells (HUASMCs) co-cultured with the blank hydrogel or NOE hydrogels containing varying contents of selenocystamine (SeCA) in the presence of S-nitrosoglutathione (GSNO) ($n = 8$). One-way analysis of variance (ANOVA) with Tukey post-hoc test was performed to determine the difference among various substrates. (* $P < 0.05$, ** $P < 0.01$, *** $P < 0.001$ and **** $P < 0.0001$).

10. Fig 5

The PCA analysis shows minimal variability and I do not see how this can be biologically relevant.

Reply: The PCA does show large variations among different groups except for the groups of Blank control and GSNO (Fig. 5a (i.e. Fig. R6)). We have discussed in the Manuscript that GSNO alone barely had any impact on the gene expression of HUASMCs. It is normal that the data of parallel experiments in each group clustered together.

Fig. R6 (i.e. Fig. 5a) | Principal component analysis (PCA) representing the general variations in gene expression of human umbilical artery smooth muscle cells (HUASMCs) among different groups.

11. Lines 460-462

316L stainless steel stents are NOT widely used in clinical practice

Reply: Thanks for the comment! We agree that bare-metal stent (BMS) of 316L is much less used in clinical practice now. However, as the first-generation BMS, it used to be widely applied in clinical practice and is still frequently used as a base platform to evaluate the performance of a coating material in animal experiment (*Biomaterials* **2016**, 87, 82; *NPG Asia Mater.* **2018**, 10, 642; *Proc. Natl. Acad. Sci. USA* **2020**, 117, 16127). To be accurate, we have deleted the words “due to its wide application in clinical practice” in the revised Manuscript.

12. Endothelial coverage

I am not clear as to how endothelial coverage was assayed for in the animals. Seems like IHC was used with antiCD31 antibodies. Was this characterized with confocal microscopy or en face imaging?

Reply: To investigate the endothelial coverage, we stained CD31 of endothelial cells on the vascular stents and then took fluorescence images of them with confocal laser scanning microscope. Afterwards, the endothelial coverage was determined by measuring the proportion of area covered by endothelial cells in each image. The average value and standard deviation of endothelial coverage were calculated with $n \geq 6$.

13. Thrombosis

I do not understand the idea of not reporting thrombosis on stents implanted in left and right iliac arteries or in the different coronary arteries of the pig. Are you saying that the placement of an uncovered stent in one iliac or coronary will have impact on the other. If this is the case for thrombosis, it should be the case for inflammation, endothelial recovery and intimal hyperplasia as well. Were there any thrombotic events in any of the animals

Reply: Sorry that we misinterpreted your meaning in the previous comments! We thought you were meaning the embolization in other organs such as lung and heart induced by the *ex vivo* thrombogenicity test. In our study, only a few cases of acute thrombosis were found during stent deployment. We have reported them in Supplementary Table 1 and Supplementary Table 3 in the revised Supplementary Information now.

14. Animal studies

Did the results vary on which coronary artery was used.

Reply: Coronary artery disease might occur to all coronary arteries. Therefore, the locations for coronary stenting in animals are usually rotated or randomized (*Ann. Biomed. Eng.* **2016**, 44, 453-465). In our study, we randomly implanted the four types of stents in three to four coronary arteries of individual pigs. Our data do not suggest that the selection of coronary artery had significant influence on the outcome.

15. Animal studies

Seems like what can be said is that NO generated coatings reduce intimal hyperplasia and have early endothelial coverage comparable to standard drug eluting stents. The NO generating coatings have greater endothelial coverage but more intimal hyperplasia early than clinical drug-

eluting stents which at three months show no difference in endothelial coverage and slightly more intimal hyperplasia.

Reply: Yes! As shown in Fig. 7d-7e in the Manuscript, the nitric oxide-eluting (NOE) hydrogel-coated stent presented complete endothelial regeneration though the area stenosis ($15.6\pm 2.7\%$) of it was slightly larger than that ($13.2\pm 2.8\%$) of DES at 2 weeks post implantation. The progress of endothelialization on DES was still incomplete even after 3 months as we noted some small regions uncovered by endothelial cells (Fig. 7f). Besides, the area stenosis ($36.4\pm 5.4\%$) of DES became significantly larger ($P < 0.01$) than that ($23.0\pm 3.1\%$) of NOE hydrogel-coated stent.

Fig. R7 | (i.e. Fig. 7d-7f) Vascular stent deployment in pigs. d, Optical images showing the cross-sections of the stented arteries after van Gieson staining. (scale bar: $500\ \mu\text{m}$) **e**, Quantitative analyses on the cross-sections ($n = 6$). **f**, Confocal laser scanning microscopy (CLSM) unveiling the endothelialization on the stents (outlined by the dashed lines). (blue: cell nucleus, green: CD31, red: F-actin). The endothelial coverages were determined for different types of stents. One-way analysis of variance (ANOVA) with Tukey post-hoc test was performed to determine the difference among various groups. ($*P < 0.05$, $**P < 0.01$, $***P < 0.001$ and $****P < 0.0001$; $#####P < 0.0001$ compared to other groups)

In short, this paper is way too long including information that distracts from the primary message – i.e. that an NO generating coating is effective in two animal models at reducing intimal hyperplasia akin to release of an intact compound. The central idea is buried in a lot of extraneous material. The paper is then simultaneously is far too long and contains too much information of no value and then does not delve into the link between potentially innovative chemistry and vascular biology. What is presented should be reduced significantly and what is not should be expanded. Entire sections could be removed AND make the paper more readable.

Reply: We agree that there are some extraneous results and discussion. To avoid distraction from the main topic, we have removed them from the revised Manuscript and Supplementary Information. For instance, the comparison between alginate-maleimide/gelatin (A-M/G) hydrogel and alginate-dialdehyde/gelatin (A-D/G) hydrogel was thoroughly removed. The key information expressed in our Manuscript is that the nitric oxide-eluting (NOE) hydrogel could not only inhibit thrombosis and repress inflammation, but also promote rapid restoration of native endothelium. These effects come from NO gas and the chemical resemblance of the hydrogel to extracellular matrix (ECM). Now the Manuscript has become more readable.

REVIEWERS' COMMENTS

Reviewer #3 (Remarks to the Author):

The authors have responded to my specific queries and though I do not agree with some statements the overall message is not affected by these (e.g. use of bare metal as controls and clinical use).

This leave us with as they now acknowledge a nitric oxide-eluting hydrogel coating that inhibits intimal hyperplasia. And now the issue is whether such a report has a novelty given the multiple reports in the literature spanning the last 20 years from Buegler et al *Coron Artery Dis*, 2000 Jun;11(4):351-7. doi: 10.1097/00019501-200006000-00009. Use of nitric-oxide-eluting polymer-coated coronary stents for prevention of restenosis in pigs, and Yoon et al 2002, *Yonsei Medical Journal* 43(2), 242-251, Local delivery of nitric oxide from an eluting stent to inhibit neointimal thickening a porcine coronary injury model, to immense work summarized nicely and in great detail in Rao et al "Nitric oxide-producing cardiovascular stent coatings for prevention of thrombosis and restenosis", *Front Bioeng Biotechnol.* 2020; 8: 578. doi: 10.3389/fbioe.2020.00578 and indeed some of the references in the submitted manuscript (e.g. ref. 32).

the authors would best be advised to reflect now placing this work into perspective within the incredible landscape of NO releasing stents.

For example, the end of Rao's paper provides a possible segue

"The progress of the NO-producing stent coatings are highly inspiring in the prevention of thrombosis and restenosis, and researchers have also attempted to improve the fabrication process and committed to developing simple materials for future clinical application. However, the immune reaction, inflammation, anaphylaxis, and biocompatibility of the coatings need to be further investigated."

REVIEWERS' COMMENTS

Reviewer #3 (Remarks to the Author):

The authors have responded to my specific queries and though I do not agree with some statements the overall message is not affected by these (e.g. use of bare metal as controls and clinical use).

This leave us with as they now acknowledge a nitric oxide-eluting hydrogel coating that inhibits intimal hyperplasia. And now the issue is whether such a report has a novelty given the multiple reports in the literature spanning the last 20 years from Buerger et al *Coron Artery Dis*, 2000 Jun;11(4):351-7. doi: 10.1097/00019501-200006000-00009. Use of nitric-oxide-eluting polymer-coated coronary stents for prevention of restenosis in pigs, and Yoon et al 2002, *Yonsei Medical Journal* 43(2), 242-251, Local delivery of nitric oxide from an eluting stent to inhibit neointimal thickening a porcine coronary injury model, to immense work summarized nicely and in great detail in Rao et al "Nitric oxide-producing cardiovascular stent coatings for prevention of thrombosis and restenosis", *Front Bioeng Biotechnol.* 2020; 8: 578. doi: 10.3389/fbioe.2020.00578 and indeed some of the references in the submitted manuscript (e.g. ref. 32).

Reply: Thanks for the comment and suggestion! The novelty of our work lies in the integration of high mechanical strength, ideal maneuverability, good biocompatibility and sustained generation of nitric oxide (NO) into a hydrogel coating. The hydrogel itself is a new material and is reported by us for the first time. Other NO-producing stents suffer from proinflammatory coatings or non-sustained generation of NO in general. For instance, in the works of Buerger *et al.* (*Coron. Artery Dis.* **2000**, 11, 351) and Yoon *et al.* (*Yonsei Med. J.* **2002**, 43, 242) as mentioned by , the authors did not find that their NO-producing stents could reduce neointimal hyperplasia (NIH) compared to the bare-metal stents and blank polymer-coated stents. They attributed such outcomes to the pro-inflammatory effects of the polymer coatings and/or lack of sustained NO generation. In contrast, our nitric oxide-eluting hydrogel coating has good biocompatibility, selectively facilitates the adhesion of endothelial cells, and persistently catalyzes NO generation. Consequently, it could effectively suppress NIH and even defeat drug-eluting stents (DES) in the long run. In addition, our hydrogel coating can be leveraged as a carrier to afford protein-based medication, which cannot be achieved by other types of coatings.

The authors would best be advised to reflect now placing this work into perspective within the incredible landscape of NO releasing stents.

For example, the end of Rao's paper provides a possible segue.

"The progress of the NO-producing stent coatings are highly inspiring in the prevention of thrombosis and restenosis, and researchers have also attempted to improve the fabrication process and committed to developing simple materials for future clinical application. However, the immune reaction, inflammation, anaphylaxis, and biocompatibility of the coatings need to be further investigated."

Reply: Thanks for the suggestion! We have further reflected the limitation and perspective of NO-releasing stents at the 'Discussion' section in the end of the revised manuscript. We also cited Rao et al.'s paper (*Front. Bioeng. Biotechnol.* **2020**, 8, 578) as Ref. 52.